# Notch signaling restricts FGF pathway activation in parapineal cells to promote their collective migration

**Lu Wei, Amir Al Oustah, Patrick Blader, Myriam Roussigné***

Centre de Biologie Intégrative (CBI), Centre de Biologie du Développement (CBD), Université de Toulouse, CNRS (UMR 5547), Toulouse, France

**Abstract** Coordinated migration of cell collectives is important during embryonic development and relies on cells integrating multiple mechanical and chemical cues. Recently, we described that focal activation of the FGF pathway promotes the migration of the parapineal in the zebrafish epithalamus. How FGF activity is restricted to leading cells in this system is, however, unclear. Here, we address the role of Notch signaling in modulating FGF activity within the parapineal. While Notch loss-of-function results in an increased number of parapineal cells activating the FGF pathway, global activation of Notch signaling decreases it; both contexts result in defects in parapineal migration and specification. Decreasing or increasing FGF signaling in a Notch loss-of-function context respectively rescues or aggravates parapineal migration defects without affecting parapineal cells specification. We propose that Notch signaling controls the migration of the parapineal through its capacity to restrict FGF pathway activation to a few leading cells.

DOI: https://doi.org/10.7554/eLife.46275.001

## Introduction

Coordinated migration of cell collectives is a widespread phenomenon, being seen predominantly during embryonic development but also during tissue repair in adults, for example. The molecular and cellular mechanisms underlying collective cell migration have been studied *in vivo* in different model organisms (*Friedl and Gilmour, 2009*; *Ochoa-Espinosa and Affolter, 2012*; *Pocha and Montell, 2014*; *Theveneau and Mayor, 2013*). Recent progress in the analysis of mechanical forces together with the development of *in vitro* models and in silico modeling have improved our understanding of coordinated cell migration. Such studies have highlighted the variability in mechanisms from one model to another, indicating that collective migration is a highly adaptive and plastic process (*Haeger et al., 2015*; *Theveneau and Linker, 2017*).

Members of the FGF family of secreted signals have been implicated in many models of cell migration. For example, FGF signaling is described to promote migration of cell collectives, potentially through chemotaxis (*Kadam et al., 2012*), through the modulation of cell adhesiveness (*Ciruna et al., 1997*; *McMahon et al., 2010*) or by increasing random cell motility (*Bénazéraf et al., 2010*). In the lateral line primordium, the FGF pathway is required for Notch-dependent formation of neuromast rosettes at the trailing edge of the migrating primordium (*Durdu et al., 2014*; *Kozlovskaja-Gumbriené et al., 2017*; *Lecaudey et al., 2008*; *Nechiporuk and Raible, 2008*) and for a leading-to-trailing signaling that prevents splitting of the primordium (*Dalle Nogare et al., 2014*), with both of these processes being required for proper lateral line primordium migration. Despite the widespread and iterative role of the FGF pathway in cell migration models, however, it is not clear how the dynamics of FGF signaling correlate with cell behaviors and how this can be modulated by other signals.

***For correspondence:**
myriam.roussigne@univ-tlse3.fr

**Competing interests:** The authors declare that no competing interests exist.

**eLife digest** Many animal cells must move through the body from the place they are born to where they are needed, for example, when embryos are developing or wounds are healing. Often cells migrate in groups, which helps them to navigate and co-ordinate more effectively. Cells typically migrate by sensing different signals across large areas, and groups of cells communicate with each other to co-ordinate their migration.

One group of cells that is studied to understand collective cell migration is found in the brain of the zebrafish. These cells are called parapineal cells and, in the developing fish, they move towards the left side of the brain under the influence of a signal called Fgf8. Although all parapineal cells can detect Fgf8, only those at the front of the migrating group respond to the signal. These are the cells that lead the migration. It was previously unclear how the capability of responding to Fgf8 is limited to only a few parapineal cells to ensure they all migrate correctly.

By studying parapineal cell migration, Wei et al. identified a different signaling system called Notch as a regulator of cell migration. When Notch signaling activity is artificially increased in the brain, parapineal cells do not respond to Fgf8 as efficiently as when levels of Notch are normal. Conversely, the number of parapineal cells that respond to Fgf8 increases when Notch signaling is lost. In both cases, migration of parapineal cells is affected. Therefore, Wei et al. showed that changing the balance of Notch signaling in the zebrafish brain modifies the ability of parapineal cells to respond to Fgf8 signal and stops parapineal cells from migrating correctly.

These results provide a general model for how cells migrate as a group. More studies are needed to see if similar mechanisms are involved in other examples of collective cell migration. This model of group migration could be applied to healthy processes such as embryonic development as well as examples of cell migration in illness such as cancer metastasis.

DOI: https://doi.org/10.7554/eLife.46275.002

The parapineal is a small group of cells that segregates from the anterior part of the pineal gland at the midline of the zebrafish epithalamus and migrates in an FGF-dependent manner to the left side of the brain (*Concha et al., 2000*; *Duboc et al., 2015*; *Roussigne et al., 2012*). To characterize the dynamics of FGF pathway activation during parapineal migration, we recently analyzed the temporal and spatial activation of a previously described FGF pathway reporter transgene, *Tg(dusp6: d2GFP)* (*Molina et al., 2007*; *Roussigné et al., 2018*). Using this reporter, we showed that the FGF pathway is activated in an Fgf8-dependant manner in only a few parapineal cells located at the migration front and that experimentally activating the FGF pathway in a few parapineal cells restores parapineal migration in *fgf8*$^{-/-}$ mutant embryos. Taken together, these findings indicate that the restricted activation of FGF signaling in the parapineal promotes the migration of the parapineal cell collective. While the parapineal can receive Fgf8 signals from both sides of the midline, focal pathway activation is primarily detected on the left (*Roussigné et al., 2018*). This asymmetry in FGF pathway activation requires the TGFβ/Nodal signaling pathway, which is activated on the left side of the epithalamus prior to parapineal migration (*Bisgrove et al., 1999*; *Concha et al., 2000*; *Liang et al., 2000*; *Roussigné et al., 2018*). Although the Nodal pathway appears to bias the focal activation of FGF signaling to the left, after a significant delay the restriction of FGF activity still occurs in the absence of Nodal activity and the parapineal migrates (*Roussigné et al., 2018*).

All parapineal cells appear competent to activate the FGF pathway begging the question as to how the activation of the pathway is restricted to only a few cells. In this study, we address whether Notch signaling might modulate the activation of FGF pathway in the parapineal. We show that while loss-of-function of Notch leads to expanded FGF pathway activation in the parapineal, activating the Notch pathway causes a strong reduction in the expression of the FGF reporter transgene, with both contexts leading to defects in parapineal migration. Loss or gain of function for Notch signaling also interferes with the specification of parapineal cell identity; loss-of-function results in a significant increase in the number of *gfi1ab* and *sox1a* expressing parapineal cells, whereas gain of Notch activity results in the opposite phenotype. In contrast, the number of parapineal cells expressing *tbx2b*, a putative marker for parapineal progenitors cells (*Snelson et al., 2008*), is not affected in either loss or gain of function for Notch. Pharmacological inhibition of Notch pathway activity

suggests that the roles of Notch in the specification and migration of parapineal cells can be uncoupled. Finally, a global decrease or an increase in the level of FGF signaling can respectively rescue or aggravate the parapineal migration defect caused by Notch loss-of-function but without affecting the specification of parapineal cells. Our data indicate that the Notch pathway regulates the specification and migration of parapineal cells independently and that the role of Notch signaling in promoting parapineal migration, but not specification, depends on its ability to restrict FGF pathway activation to a few parapineal cells.

## Results

### The parapineal of *mindbomb* mutant embryos display expanded FGF pathway activation

In models of cell migration during sprouting of tubular tissues, Notch-Delta mediated cell-cell communication contributes to tip cell selection by restricting the ability of followers cells to activate RTK signaling (*Ghabrial and Krasnow, 2006*; *Ikeya and Hayashi, 1999*; *Riahi et al., 2015*; *Siekmann and Lawson, 2007a*). To address whether Notch signaling could similarly restrict FGF pathway activation in the freely moving group of parapineal cells, and thus promote its migration, we analyzed the expression of an FGF pathway activity reporter transgene, *Tg(dusp6:d2EGFP)* (*Molina et al., 2007*), in embryos mutant for the *mindbomb* (*mib*$^{ta52b}$) gene, a well-described loss-of-function context for the Notch pathway (*Itoh et al., 2003*). At 32 hours post-fertilization (hpf), we observed a larger number of *Tg(dusp6:d2EGFP)* expressing cells in the parapineal of *mib*$^{-/-}$ mutant embryos (7 ± 3 d2EGFP+ cells) compared to siblings (4 ± 2 d2EGFP+ cells; p-value=3.845e-05) (*Figure 1A–1B' and E*). While the number of *Tg(dusp6:d2EGFP)*+ cells increases, we found that the mean intensity of *Tg(dusp6:d2EGFP)* does not change significantly in *mib*$^{-/-}$ mutants indicating that it is the restriction of FGF activation to few parapineal cells rather than the level of FGF activity that is affected in *mib*$^{-/-}$ mutants (*Figure 1—figure supplement 1*). The increase in *Tg(dusp6:d2EGFP)* expressing cells was not accompanied by an increase in the number of parapineal cells expressing *sox1a*, the earliest described parapineal specific marker (detected from 28 hpf in the parapineal, *Clanton et al., 2013*) (*Figure 1C–1D and G*), although we observed a slight increase in the total number of parapineal cells as determined using nuclear staining to visualize the parapineal rosette (22 ± 7 in *mib*$^{-/-}$ mutants compared to 19 ± 5 in sibling control embryos; p-value=0.037) (*Figure 1F*).

To confirm that the phenotypes observed in *mib*$^{-/-}$ mutants are caused by Notch pathway loss-of-function, we analyzed the parapineal in embryos injected with a morpholino (MO; antisense oligonucleotide) blocking translation of the two zebrafish orthologs of *su(H)/rbpj* (*Echeverri and Oates, 2007*); Rpbj proteins are transcription factors required for canonical Notch signaling (*Fortini and Artavanis-Tsakonas, 1994*; *Hsieh et al., 1996*). As observed in *mib*$^{-/-}$ mutants, the proportion of *Tg(dusp6:d2EGFP)* expressing cells was increased in the parapineal of embryos injected with *rbpja/b* MO (4 ng) (10 ± 6 d2EGFP+ cells) compared to controls (6 ± 4 d2EGFP+ cells; p-value=0,0016) (*Figure 1—figure supplement 2, C*). Altogether, our results indicate that the FGF signaling pathway is activated in more parapineal cells when Notch signaling is abrogated.

### Components of the Notch pathway are expressed in parapineal cells

To address whether the Notch pathway is active in the parapineal, we checked the expression of a well-characterized Notch reporter transgene, *Tg(Tp1bglob:EGFP)* (*Corallo et al., 2013*; *Parsons et al., 2009*), in the epithalamus at 30 hpf. Although rare, cells expressing GFP could be detected in the parapineal at 30 hpf (n = 3/19; *Figure 1—figure supplement 3, A–B'*).

In parallel, we analyzed the expression of genes encoding Notch ligands. We found that *deltaB* is expressed in one or two parapineal cells at both 28 hpf (n = 6/12) and 32 hpf (n = 17/23) (*Figure 1—figure supplement 3, C–D'*); no expression could be detected in the parapineal of the remaining embryos (n = 6/12 at 28 hpf or n = 6/23 at 32 hpf), suggesting that *deltaB* expression is highly dynamic. We found that *deltaA* mRNA could also be detected at 32 hpf in 1–2 parapineal cell, although its expression is less robust than *deltaB* (n = 3/10) (not shown). When detected in the parapineal, *deltaB* expression was found to overlap with *Tg(dusp6:d2EGFP)* expression in between half (n = 3/6 at 28 hpf) and two thirds of the embryos (n = 11/17 at 32 hpf).

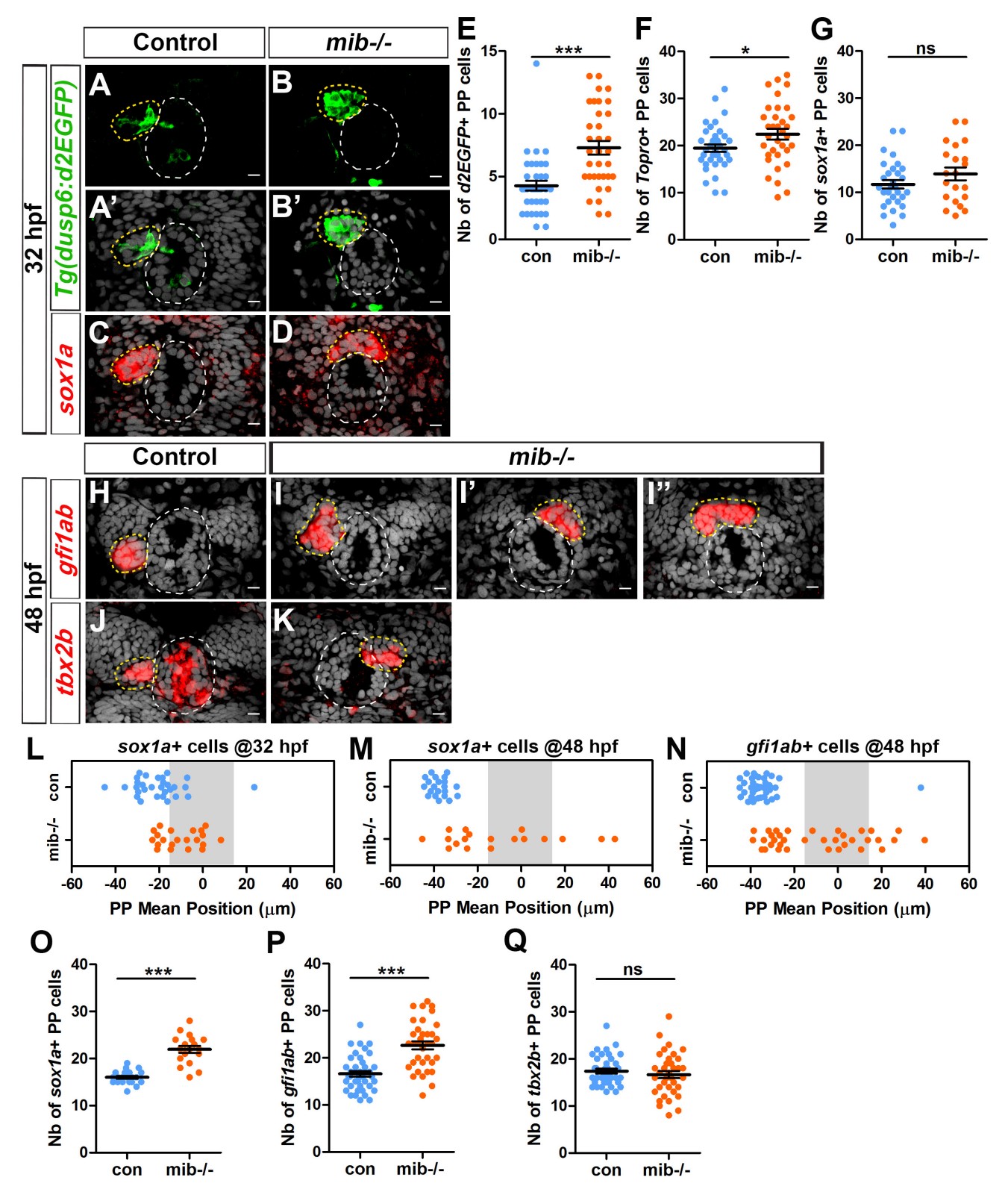

**Figure 1.** Increased activation of FGF signaling in *mindbomb* mutants correlates with defects in migration and specification of parapineal cells. (A–D) Confocal sections showing the expression of the *Tg(dusp6:d2EGFP)* transgene (Green, (A–B') or *sox1a* (red, (C–D) in the epithalamia of 32 hpf in control (A, A', n = 36 and C, n = 30) and in *mib*[-/-] mutant embryos (B, B', n = 34 and D, n = 21). Confocal sections are merged with a nuclear staining (gray, A', B', C, D) that makes the epiphysis (white circle) and parapineal (yellow circle) visible. Embryo view is dorsal, anterior is up; scale bar = 10 µm. (E–G) Dot
*Figure 1 continued on next page*

*Figure 1 continued*

plots showing the number (**E–G**) of *Tg(dusp6:d2EGFP)* (**E**), Topro-3 (**F**) and *sox1a* (**G**) positive parapineal cells in control (blue dots) or *mib*$^{-/-}$ mutant embryos (orange dots) at 32 hpf, with mean ± SEM. *Tg(dusp6:d2EGFP)* FGF reporter is expressed in more parapineal cells in *mib*$^{-/-}$ than controls (**E**; 7 ± 3 d2EGFP + cells in *mib*$^{-/-}$ mutant compared 4 ± 2 in siblings; p-value=3.845e-05, Welch t-test) while the expression of *sox1a* is similar in both contexts (**G**); the average number of parapineal cells counted with a nuclear marker increases slightly (**F**; 22 ± 7 in *mib*$^{-/-}$ mutants versus 19 ± 5 in sibling control embryos; p-value=0.037, Welch t-test). (**H–I''**) Confocal sections showing the expression of *gfi1ab* (red) at 48 hpf in control embryos (**H**; n = 40) and in three examples of *mib*$^{-/-}$ mutant embryos (**I-I''**; n = 35). (**J–K**) Confocal sections showing the expression of *tbx2b* (red) at 48 hpf in control embryos (**J**; n = 39) and in one example of *mib*$^{-/-}$ mutants embryo with the parapineal on the right (**K**; n = 36). (**L–N**) Dot plots showing the mean position of parapineal cells expressing *sox1a* at 32 hpf (**L**) *sox1a* at 48 hpf (**M**) or *gfi1ab* (**N**) (in μm distance to the brain midline (x = 0)) with each dot representing an embryo. Parapineal migration is usually delayed in *mib*$^{-/-}$ mutants at 32 hpf (**L**). At 48 hpf, the parapineal of *mib*$^{-/-}$ mutant embryos either did not migrate (n = 12/35 with a parapineal mean position between −15 μM and +15 μM (shaded zone) relative to brain midline (Reference 0)) or migrates either to the left (n = 17/35) or to the right (n = 6/35) (**N**, orange dots), while it usually migrated to the left in control embryos (**N**, n = 39/40, blue dots); p-value<0.0001, Welch t-test on absolute values. (**O–Q**) Number of parapineal cells expressing *sox1a* (**O**), *gfi1ab* (**P**) and *tbx2b* (**Q**) at 48 hpf in control (blue dots) or in *mib*$^{-/-}$ mutant embryos (orange dots). The number of *sox1a* and *gfi1ab+* positive parapineal cells at 48 hpf is increased in *mib*$^{-/-}$ mutant embryos (p-value<0.0001 in Welch t-test) compared with controls (**O, P**) while the number of *tbx2b* expressing cells is unchanged (**Q**). Mean ± SEM are indicated as long and short bars. *** p-value<0.0001; * p-value<0.05 in Welch t-test. Data are representative of three experiments (**H–I''**, **N, P**) or two independent experiments (**A–G, J–M, O, Q**). See also *Figure 1—figure supplement 1*, *Figure 1—figure supplement 2* and *Figure 1—figure supplement 3*. Source files used for dot plots and statistical analysis are available in *Figure 1—source data 1*.
DOI: https://doi.org/10.7554/eLife.46275.003

The following source data and figure supplements are available for figure 1:

**Source data 1.** Source files for data used to generate dot plots in *Figure 1*.
DOI: https://doi.org/10.7554/eLife.46275.007
**Figure supplement 1.** The number but not the mean intensity of *Tg(dusp6:d2EGFP)* expressing cells is increased in *mib*$^{-/-}$ mutants.
DOI: https://doi.org/10.7554/eLife.46275.004
**Figure supplement 2.** *rbpja/b* morphants phenocopy *mindbomb* mutants.
DOI: https://doi.org/10.7554/eLife.46275.005
**Figure supplement 3.** Components of the Notch pathway are mosaically expressed in the parapineal.
DOI: https://doi.org/10.7554/eLife.46275.006

Therefore, although the expression of *Tg(Tp1:EGFP)* Notch reporter transgene is not robust in the parapineal, the mosaic expression of *deltaA* and *deltaB* ligands in a few parapineal cells supports the existence of Notch signaling activity within the parapineal.

## Loss of Notch signaling results in defects in parapineal migration

Interestingly, while in controls the parapineal had usually initiated migration at 32 hpf, it was still detected at the midline in most stage-matched *mib*$^{-/-}$ mutant embryos (*Figure 1A–D and L*). To address the effect of Notch inhibition on parapineal migration further, we examined *mib*$^{-/-}$ mutant embryos at a later stage, when the parapineal had unambiguously migrated in all controls. Analyzing *sox1a* (*Clanton et al., 2013*) expression at 2 days post-fertilization revealed that the parapineal failed to migrate in about a third of *mib*$^{-/-}$ embryos (defined by parapineal mean position within −15 μm and +15 μm of the midline (gray-shaded zone)) (*Figure 1M*). We confirmed this result by using another marker, *gfi1ab* (*Dufourcq et al., 2004*), whose expression is detected at a later stage (from 36 to 40 hpf) than *sox1a* (from 28 hpf) in parapineal cells. Using *gfi1ab*, we found that the parapineal had not migrated in 35% of *mib*$^{-/-}$ embryos (n = 12 of 35 embryos; *Figure 1I'' and N*), while it migrated to the left in more than 95% of control embryos (n = 39/40, *Figure 1H and N* p-value<0.0001); the parapineal in *mib*$^{-/-}$ mutant embryos also migrated to the right more often than in controls (17%, n = 6/35; *Figure 1I' and N*).

In embryos injected with *rbpja/b* MO (4 ng), the mean position of parapineal cells at 32 hpf was closer to the midline, suggesting a similar defect in parapineal migration as observed in *mib*$^{-/-}$ mutant embryos (*Figure 1—figure supplement 2, D*). Using *gfi1ab* as a marker at 48 hpf, we found that both parapineal migration *per se* and the orientation of migration were significantly affected in *rbpja/b* morphants (*Figure 1—figure supplement 2, A–B', E*). While the parapineal migrated to the left in most uninjected controls (94%, n = 31/33), in *rbpja/b* morphants migration was blocked (13% of the embryos, n = 6/46; p-value=0.0001) or its orientation was partially randomized (left in 67% of the embryos, n = 31/46; right in 20% of the embryos, n = 9/46; p-value=0.0002) (*Figure 1—figure supplement 2, A–B', E*).

Taken together, our data show that loss of Notch signaling results in defects both in parapineal migration *per se* (distance from the midline) and in the laterality of migration (left orientation), and that this correlates with the FGF signaling pathway being activated in more parapineal cells.

## Loss of Notch signaling results in an increase in the number of certain parapineal cell subtypes

While quantifying parapineal mean position at 48 hpf, we observed an overall increased in the number of *sox1a* expressing cells in *mib*$^{-/-}$ mutants at this stage (*Figure 1O*) despite finding no significant change at 32 hpf (*Figure 1G*). Similarly, we found that the number of *gfi1ab*-positive cells at 48 hpf was significantly increased in *mib*$^{-/-}$ mutants (23 ± 5) compared to control embryos (17 ± 4; p-value=3.901e-07) (*Figure 1H–1I'' and P*). In embryos injected with 4 ng of *rbpj a/b* MO, we also observed that the number of *gfi1ab* expressing parapineal cells at 48 hpf was significantly increased (19 ± 6) compared to control embryos (14 ± 3; p-value<0.0001) (*Figure 1—figure supplement 2, A–B', F*). In contrast, the number of parapineal cells expressing *tbx2b*, a parapineal marker previously suggested to be required for the specification of parapineal cells (*Snelson et al., 2008*), was not increased in *mib*$^{-/-}$ mutants (*Figure 1J–1K and Q*).

Taken together, our data show that loss of Notch signaling results in defects in parapineal migration and, at later stages, an increase in the number of *gfi1ab* and *sox1a* expressing parapineal cells (putative differentiated parapineal cells) while the number of *tbx2b* expressing cells (putative parapineal progenitors) is unchanged.

## The roles of Notch signaling in the specification and migration of parapineal cells can be uncoupled

Our data show that blocking Notch signaling leads to an expansion of FGF pathway activation in the parapineal, defects in parapineal migration and laterality, and an increase in the number of *gfi1ab* and *sox1a* expressing parapineal cells. With the aim of unraveling potential causative links between these different phenotypes, we used a pharmacological inhibitor of the γ-secretase complex, to block the Notch signaling pathway during different time windows (*Romero-Carvajal et al., 2015*; *Rothenaigner et al., 2011*); γ-secretase activity is required for the release of the intracellular domain of Notch, NICD, during activation of the canonical pathway (*Geling et al., 2002*).

We first treated wild-type embryos with LY411575 between 22 and 32 hpf, a time window corresponding to parapineal segregation from the pineal gland and the onset of its migration. While no change in *gfi1ab* expression was detected in embryos treated with 30 µM LY411575 (*Figure 2C*), a higher concentration of LY411575 (100 µM) resulted in an increase in the number of *gfi1ab*-positive cells in treated embryos (*Figure 2A–2C*); neither treatment resulted in defects in parapineal migration (*Figure 2D*). As this effect of LY411575 treatment was modest, we next treated embryos heterozygous for *mib* mutation (*mib*$^{+/-}$) during the same time window thinking that this might provide a sensitized background for the drug. *mib*$^{+/-}$ embryos treated with the lower dose of LY411575 show a strong increase in the number of *gfi1ab* expressing parapineal cells (28 ± 5) compared to LY411575 treated wild-type controls (17 ± 3; p-value=1.0e-10) or DMSO-treated *mib*$^{+/-}$ (20 ± 2; p-value=2.2e-10) (*Figure 2E–2H and I*). As before, however, we did not detect a parapineal migration defect in LY411575-treated *mib*$^{+/-}$ (*Figure 2E–2H and J*), even when we increased the dose of LY411575 to 200 µM (*Figure 2—figure supplement 1*). Therefore, although LY411575 treatment from 22 to 32hpf can synergize with a *mib*$^{+/-}$ genetic background to promote an increase in the number of *gfi1ab*-positive parapineal cells, it does not affect parapineal migration. These data indicate that the role of Notch in controlling the specification of parapineal cells can be uncoupled from its function in parapineal migration.

Finally, we saw no significant increase in the number of *Tg(dusp6:d2EGFP)* positive cells in *mib*$^{+/-}$ embryos treated with LY411575 during the 22 to 32 hpf time window (*Figure 2K*). This wild-type level of FGF pathway activation correlates with correct migration of the parapineal in LY411575-treated embryos and is consistent with our previous results suggesting that restricted FGF pathway activation is important for correct parapineal migration (*Roussigné et al., 2018*).

As the parapineal has usually started to migrate by 32 hpf, it is unlikely that the full parapineal migration defects observed in some *mib*$^{-/-}$ mutants is a consequence of a role of the Notch pathway after this stage; Notch signaling could nevertheless contribute to migration after 32 hpf. To address

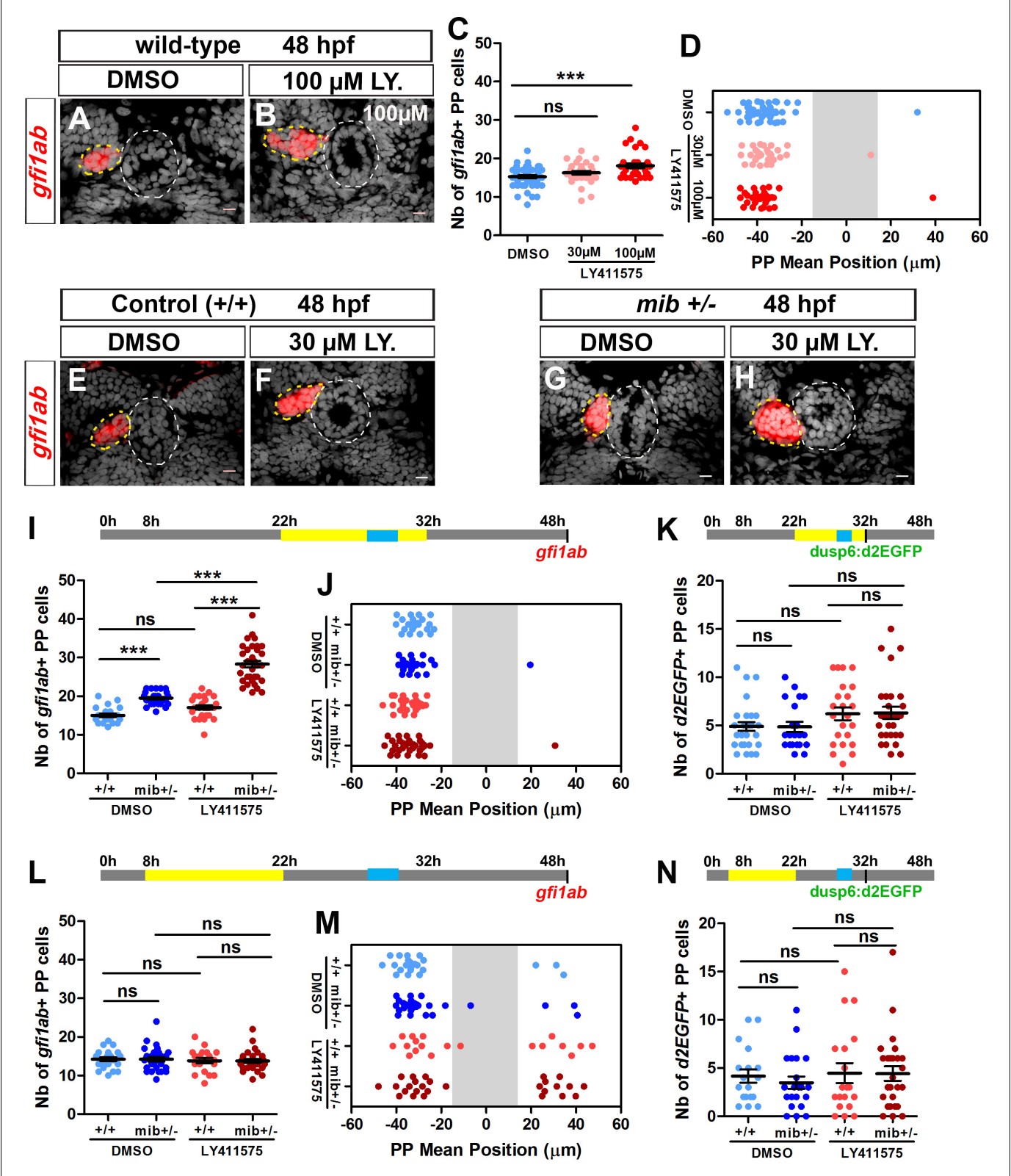

**Figure 2.** Uncoupled roles for the Notch pathway in the specification, laterality and migration of parapineal cells. (A–B) Confocal sections showing the expression of *gfi1ab* (red) in embryos treated with DMSO (A; n = 24) or 100 µM LY411575 (B; n = 32) from 22 to 32 hpf and fixed at 48 hpf, merged with nuclear staining (gray). Embryo view is dorsal, anterior is up; epiphysis (white circle) and parapineal (yellow circle); scale bar = 10 µm. (C–D) Dot plots showing the number (C) and the mean position (D) of *gfi1ab* expressing parapineal (PP) cells at 48 hpf in embryos treated with DMSO (controls, blue

*Figure 2 continued on next page*

Figure 2 continued

dots, n = 47), with 30 µM LY411575 (light red dots, n = 31) or 100 µM LY411575 (red dots, n = 32) from 22 hpf to 32 hpf, with mean ± SEM; *** p-value=0.0003. (E–H) Confocal sections showing the expression of *gfi1ab* (red) merged with nuclear staining (gray) at 48 hpf, in wild-type (+/+) (E, F) or *mib*[+/-] embryos (G, H) treated with DMSO (E, n = 23 or G, n = 26) or with LY411575 (F, n = 25 and H, n = 34) from 22 to 32 hpf. (I–N) Upper panels show a schematic of the LY411575 treatment timeline (yellow box, 22 to 32 hpf for I-K or 8 to 22 hfp for dot plots L-N), and the time window corresponding to when the parapineal initiates its migration (blue box, 28 to 30 hpf). Dot plots showing the number (I, L) and the mean position (J, M) of *gfi1ab* expressing cells at 48 hpf, or the number of *Tg(dusp6:d2EGFP)* expressing cells at 32 hpf (K, N), in the parapineal of DMSO-treated wild-type (+/+, light blue dots; I-J, n = 23; L-M, n = 12; K, n = 28; N, n = 18), DMSO-treated *mib*[+/-] heterozygote (dark blue dots; I-J, ± = 26; L-M, n = 17; K, n = 21; N, n = 21), LY411575-treated wild-type ( light red dots; I-J, n = 25; L-M, n = 11; K, n = 23, N, n = 19) and LY411575-treated *mib*[+/-] (dark red dots, I-J, n = 34 or L-M, n = 16; K, n = 29, N, n = 26); each dot represents a single embryo. Mean ± SEM is shown in I, K, L, N; *** p-value<0.0001, in Wilcoxon test and Welch t-test. In J and M, there is no defect in migration *per se* (ns p-value in Welch t-test on absolute value) but LY411575 treatment from 8 to 22 hpf (M) triggers a laterality defect (increased number of embryos with a parapineal on the right). Data are representative of three (E–H, I–K) or two experiments (A–D and L–N). See also *Figure 2—figure supplement 1*, *Figure 2—figure supplement 2* and *Figure 2—figure supplement 3*. Source files used for dot plots and statistical analysis are available in *Figure 2—source data 1*.

DOI: https://doi.org/10.7554/eLife.46275.008

The following source data and figure supplements are available for figure 2:

**Source data 1.** Source files for data used to generate dot plots in *Figure 2*.
DOI: https://doi.org/10.7554/eLife.46275.012
**Figure supplement 1.** Effect of treatment with high dose of γ-secretase inhibitor from 22 to 32 hpf on the specification and migration of parapineal cells.
DOI: https://doi.org/10.7554/eLife.46275.009
**Figure supplement 2.** LY411575 treatment after 32 hpf does not affect the migration or the specification of parapineal cells.
DOI: https://doi.org/10.7554/eLife.46275.010
**Figure supplement 3.** Early Notch loss-of-function results in bilateral Nodal pathway activation in the epithalamus.
DOI: https://doi.org/10.7554/eLife.46275.011

this possibility, we treated embryos with LY411575 during a later (32–36 hpf) or extended time window (22–48 hpf). We did not observe any effect of these treatments on the migration as assessed by *gfi1ab* expression at 48 hpf (*Figure 2—figure supplement 2, A,B*). Likewise, we did not observe changes in the number of *gfi1ab+* cells upon these late treatments (*Figure 2—figure supplement 2, C,D*).

## Notch activity is required early for unilateral activation of the Nodal pathway in the epithalamus

To determine whether the parapineal migration defects we observed in *mib*[-/-] mutant embryos and *rbpja/b* morphants might be an indirect consequence of an earlier role of Notch signaling, we also analyzed the epithalamus of embryos treated with LY411575 during an earlier 8 to 22 hpf time window. This early drug treatment did not interfere with the number of *gfi1ab*-positive cells (*Figure 2L*), or with migration in itself (*Figure 2M*), but led to a partial randomization of the direction of parapineal migration as shown by a significant increase in the number of embryos with a right parapineal (*Figure 2M*).

We hypothesized that the partial randomization of parapineal sidedness observed in *mib*[-/-] mutants, *rbpja/b* morphants or in embryos treated with LY411575 from 8 to 22 hpf could be caused by changes in the activation pattern of Nodal signaling in the epithalamus (*Concha et al., 2000*; *Liang et al., 2000*; *Regan et al., 2009*). To address this possibility, we analyzed the expression of *pitx2c*, a Nodal signaling target gene (*Concha et al., 2000*; *Essner et al., 2000*; *Liang et al., 2000*), in the different contexts of Notch loss-of-function. While *pitx2c* expression is detected in the left epithalamus in control embryos between 28 and 32 hpf (n = 26/28), we observed that its expression is bilateral in most *mib*[-/-] mutant embryos (n = 26/29) (*Figure 2—figure supplement 3, A–C,D*), in a majority of *rbpja/b* morphants (n = 13/23, *Figure 2—figure supplement 3, E*) and in approximately half of the embryos treated with LY411575 from 8 to 22 hpf, regardless of whether they were heterozygotes for the *mib* mutation (*mib*[+/-], n = 6/10) or not (n = 4/13) (*Figure 2—figure supplement 3, G*). In contrast, the expression of *pitx2c* was indistinguishable from controls in embryos treated with LY411575 during the later time window (22–32 hpf, *Figure 2—figure supplement 3, H*) or in embryos expressing NICD from 26 hpf (*Figure 2—figure supplement 3, F*), which was expected given the absence of laterality defects in these contexts.

Despite triggering defects in parapineal laterality, the early time window of LY411575 treatment (8 to 22 hpf) did not affect parapineal migration *per se* and, as observed for the late time window (22–32 hpf), this correlates with no significant change in the *Tg(dusp6:d2EGFP)* expression pattern (*Figure 2N*). Our data indicate that the partial randomization of parapineal migration observed in *mib*[-/-] mutants or *rbpj* morphants is due to an early role of Notch pathway in restricting Nodal signaling to the left epithalamus and not to changes in the pattern of FGF activation.

## Notch gain of function inhibits FGF pathway activation in the parapineal, and blocks the specification and migration of parapineal cells

When we abrogated Notch signaling, we observed an increase in the number of *Tg(dusp6:d2EGFP)* + parapineal cells and correlated defects in parapineal migration (*Figure 1E and L–M*, *Figure 1— figure supplement 2, C–E*). To address further the role of the Notch pathway in modulating FGF activation, we analyzed the phenotypes associated with global activation of Notch signaling. For this, we used previously described transgenic lines, *Tg(hsp70:gal4)* and *Tg(UAS:NICD-myc)* (*Scheer and Campos-Ortega, 1999*), to induce widespread expression of the Notch Intracellular Domain (NICD) upon heat shock. In most embryos globally expressing NICD from 26 hpf, we observed a strong decrease in the number of *Tg(dusp6:d2EGFP)* expressing parapineal cells at 36 hpf (2.5 ± 2 cells) compared with the control embryos (5 ± 4; p-value=0.01) (*Figure 3A–3B and G*); the mean intensity of d2EGFP fluorescence was also significantly decreased in these embryos compared with the controls (p-value=0.0001) (*Figure 3A–3B and H*).

To assess a potential correlation between the inhibition of FGF pathway activation and defects in parapineal migration, we sought to analyze the mean position of *sox1a* or *gfi1ab* expressing cells at 36 or 48 hpf as we had done for Notch loss-of-function embryos. However, following heat shock at 26 hpf, *sox1a* expression was lost in most 36 hpf embryos (*Figure 3C–D and I*) and was strongly decreased at 48 hpf (*Figure 3M–3N and S*). Similarly, although the number of *gfi1ab* positive cells did not vary significantly in *Tg(hsp70:gal4); Tg(UAS:NICD-myc)* embryos heat shocked at 22 and 24 hpf (*Figure 3—figure supplement 1, A–D,I–J*), it was strongly decreased in embryos expressing NICD beginning from 26, 28 and 32 hpf (*Figure 3O–P and T* and *Figure 3—figure supplement 1E– H,K–L*); in embryos heat shocked at 26 hpf or 28 hpf, *gfi1ab* staining was often completely lost in the parapineal (n = 7/25 or n = 8/26, respectively) or detected in less than 4 cells (n = 11/25 or n = 14/26, respectively) (*Figure 3O–P and T* and *Figure 3—figure supplement 1, E–F,K*). However, nuclear staining indicated that the parapineal rosettes can be detected in most of the embryos expressing NICD (*Figure 3A–F and M–R*, yellow circle), suggesting that the parapineal does form despite global activation of Notch. Moreover, *tbx2b* expression was not affected in NICD expressing embryos either at 36 hpf (*Figure 3E–F and J*) or at 48 hpf (*Figure 3Q–R and U*). This result suggests that global Notch activation inhibits the specification/differentiation of *gfi1ab*+ or *sox1a*+ cells from *tbx2b*+ progenitors.

As NICD expression led to a loss of *Tg(dusp6:d2EGFP)* and *sox1a* expression, we first relied on nuclear staining to assess parapineal cells mean position in this context at 36 hpf. We found that the parapineal was closer to the midline in embryos expressing NICD from 26 hpf, (−8.7 ± 6.3 μM; n = 16) compared to control embryos (−20.5 ± 8.3 μM; n = 16; p-value=0.001) (*Figure 3A–F and K*). This migration defect was also revealed by analyzing the mean position of *tbx2b* expressing parapineal cells at 36 hpf (p-value=0.0017) (*Figure 3L*). Defects in parapineal migration were confirmed at 48 hpf using *sox1a*, *gfi1ab* and *tbx2b* as markers to assess PP mean position (*Figure 3M–R and V– X*); for instance, when *gfi1ab*-positive cells were detected, their mean position was significantly closer to the midline in embryos expressing NICD from 26 hpf (−10.1 ± 14 μM; n = 18) compared to control embryos (−34.6 ± 7.5 μM; n = 27) (p-value<0.0001) (*Figure 3O–P and W*). Parapineal mean position was also affected in embryos with NICD induced just before parapineal formation (heat shock at 22 and 24 hpf), or after parapineal formation (heat shock at 28 and 32 hpf), although in the latter case, the penetrance varies (*Figure 3—figure supplement 1, A–H,M–P*).

Altogether, our data show that global activation of Notch signaling inhibits the migration of parapineal cells, and that this is correlated with a decrease in the level of FGF signaling detected in the parapineal. Ectopic Notch signaling also decreases the number of *gfi1ab*+ and *sox1a*+ parapineal cells, a phenotype opposite to that observed in Notch loss-of-function contexts.

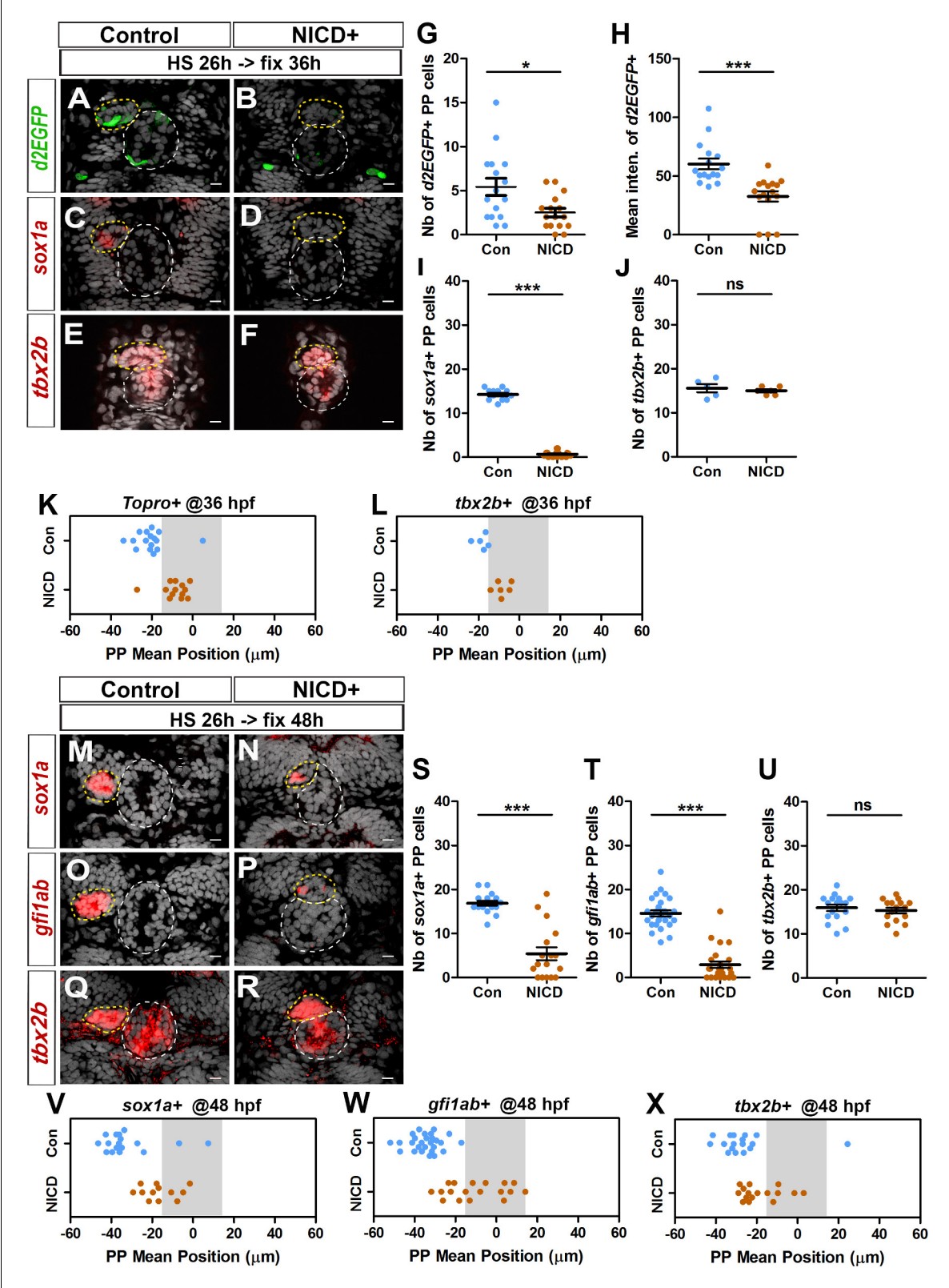

**Figure 3.** Ectopic Notch signaling triggers decreased FGF activation and defects in migration and specification of parapineal cells. (A–F) Confocal sections showing the expression of *Tg(dusp6 :d2EGFP)* (A-B, green), *sox1a* (C-D, red) or *tbx2b* (E-F, red) at 36 hpf, in control embryos (A, n = 16; C, n = 12; E, n = 5) or in *Tg(hsp70l:Gal4), Tg(UAS:myc-notch1a-intra)* embryos (B, n = 16; D, n = 17; F, n = 6) following a heat-shock (HS) at 26 hpf; sections are merged with nuclei staining (gray). (G–L) Dot plots showing the number (G) and the mean intensity fluorescence (H) of *Tg(dusp6 :d2EGFP)*

*Figure 3 continued on next page*

*Figure 3 continued*

expressing parapineal cells, the number of *sox1a* (I) and *tbx2b* (J) expressing parapineal cells, or the mean position of parapineal cells highlighted by Topro-3 nuclei staining (K) and *tbx2b* + parapineal cells (L) in controls (blue dots) or in NICD expressing embryos (orange dots) at 36 hpf following heat shock at 26 hpf. (M–R) Confocal sections showing the expression of *sox1a* (M–N), *gfi1ab* (O–P) or *tbx2b* (Q–R) (red) merged with nuclei staining (gray), at 48 hpf, in the epithalamia of control (M, n = 17; O, n = 27; Q, n = 17) or *Tg(hsp70l:Gal4);Tg(UAS:myc-notch1a-intra)* double transgenic embryos (N, n = 17; P, n = 25; R, n = 16), following heat-shock (HS) at 26 hpf. The expression of *sox1a* and *gfi1ab* is lost or decreased while *tbx2b* expression is unchanged in the parapineal of NICD expressing embryos. (S–X) Dot plots showing the number of *sox1a* (S), *gfi1ab* (T) and *tbx2b* (U) expressing parapineal cells at 48 hpf in controls (blue dots) or in embryos expressing NICD after heat shock at 26 hpf (orange dots) and the corresponding mean position of the cells (V–X) when expression was detected (number of *sox1a* + or *gfi1ab*+ cells > 0). In confocal sections, embryo view is dorsal, anterior is up; epiphysis (white circle) and parapineal gland (yellow circle); scale bar = 10 µm. Mean ± SEM is indicated on dot plots G-J and S-U; *** p-value<0.0001, * p-value<0.05 in Welch t-test and Wilcoxon test. For migration dot plots, p-value<0.01 (L) or p-value<0.001 (K and V–X) in pairwise Wilcoxon test and Welch t-test on absolute values. Data are representative of three (O, P, T, W) or two experiments (A–D, G–I, M–N, Q–R, S, U, V, X); data based on *tbx2b* expression at 36 hpf (E–F, J, L) represents one experiment. See also *Figure 3—figure supplement 1*. Source files used to generate dot plots and for statistical analysis are available in *Figure 3—source data 1*.

DOI: https://doi.org/10.7554/eLife.46275.013

The following source data and figure supplement are available for figure 3:

**Source data 1.** Source files for data used to generate dot plots in *Figure 3*.

DOI: https://doi.org/10.7554/eLife.46275.015

**Figure supplement 1.** Effect of activation of the Notch pathway at 22, 24, 28 and 32 hpf on the migration and specification of parapineal cells.

DOI: https://doi.org/10.7554/eLife.46275.014

## Decreasing FGF signaling rescues the parapineal migration defects in loss of Notch context while increasing FGF signaling aggravates it

Inhibiting or activating the Notch pathway results in reciprocal effects on FGF pathway activation in the parapineal, as seen by an increase or decrease in the number of *Tg(dusp6:d2EGFP)* expressing parapineal cells, respectively. Both contexts are also associated with defects in parapineal migration, suggesting that Notch-dependent control of FGF activation in parapineal cells is important for their collective migration. To investigate the link between Notch and FGF signaling further, we analyzed whether the migration phenotype observed in *mib*[-/-] mutants could be rescued by decreasing FGF signaling, using a pharmacological inhibitor of the FGF pathway (*Mohammadi et al., 1997*). We have previously shown that treating wild-type embryos with 10 µM SU5402 interferes with parapineal migration (*Regan et al., 2009*) and with the expression of the *Tg(dusp6:d2EGFP)* FGF reporter (*Roussigné et al., 2018*). Treating embryos with 5 µM SU5402, however, does not affect parapineal migration (*Figure 4A–4B and F*); the parapineal migrates in all SU5402 treated embryos (n = 31/31) as well as in all DMSO-treated control embryos (n = 32/32) (*Figure 4F*, purple versus blue dots). Using this suboptimal dose, parapineal migration was partially rescued in *mib*[-/-] embryos, with 19% of SU5402-treated *mib*[-/-] embryos showing mean parapineal position between −15 and +15 µm (n = 8/41) compared to 52% of DMSO treated *mib*[-/-] embryos (n = 16/31) (*Figure 4C–4D and F*, yellow versus orange dots; p-value=0.01). Thus, decreasing the level of FGF signaling activity can partially restore parapineal migration in a context where FGF activation is expanded supporting the hypothesis that Notch promotes parapineal migration through restricting FGF pathway activation.

Suboptimal SU5402 treatment had no effect on parapineal cells specification, either in *mib*[-/-] mutant embryos or siblings (*Figure 4E*). As previously observed, we detected an increase in the number of *gfi1ab* expressing parapineal cells in DMSO-treated *mib*[-/-] embryos (18 ± 5) compared to controls (15 ± 2; p-value=0.0042). The number of *gfi1ab* positive cells at 48 hpf was also increased in SU5402 treated *mib*[-/-] embryos (19 ± 7) compared to SU5402-treated controls (15 ± 3; p-value=0.0086); however, there was no significant difference in the number of *gfi1ab* expressing cells between DMSO-treated *mib*[-/-] embryos and SU5402 treated *mib*[-/-] embryos (p-value=0.5687). These data support further that the role of Notch signaling in controling the number of parapineal cells is independent from its function in parapineal migration and does not involve Notch mediated modulation of FGF signaling.

To address the connection between Notch and FGF signaling in an alternative way, we asked whether ectopic activation of the FGF pathway could elicit a more severe phenotype in a loss-of-function context for Notch. To achieve this, we used a transgenic line that expressed a constitutively activated Fgf receptor after heat shock, *Tg(hsp70l:Xla.Fgfr1,cryaa:DsRed)* (*Marques et al., 2008*), in

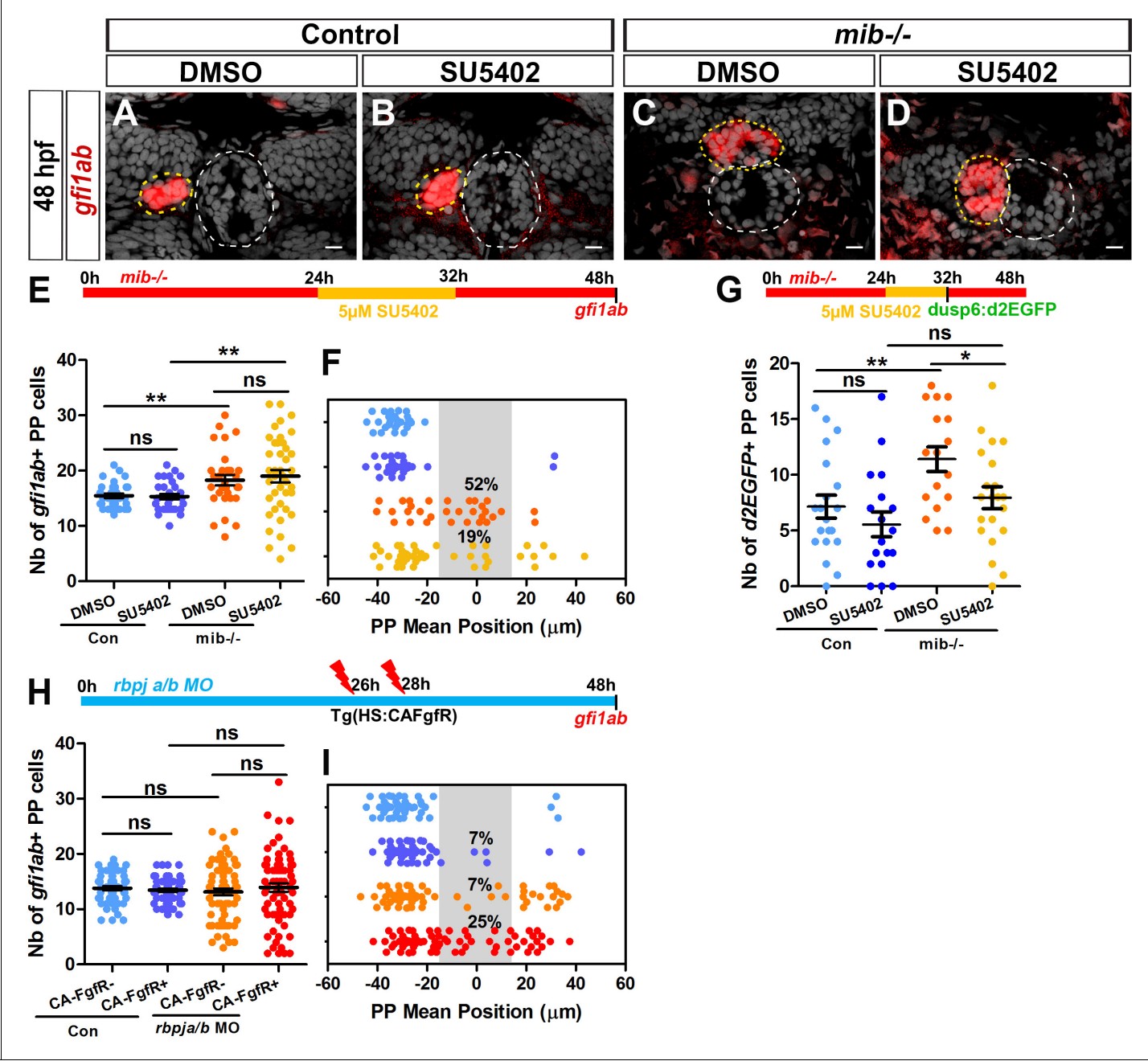

**Figure 4.** Decreasing or increasing FGF signaling rescues or aggravates the parapineal migration defect in Notch loss-of-function. (A–D) Confocal sections showing the expression of *gfi1ab* (red) merged with nuclei staining (gray) at 48 hpf in representative control sibling (A–B) or *mib*-/- mutant embryos (C–D) treated from 24 to 32 hpf with DMSO (A, C) or 5 µM SU5402 (B, D). Parapineal migration is not affected in controls embryos treated with 5 µM of SU5402 (A–B); in C-D, examples show a parapineal that failed to migrate in a DMSO treated *mib*-/- mutant embryos (C) or that migrated to the left in a SU5402 treated *mib*-/- mutant embryos (D). Embryo view is dorsal, anterior is up; epiphysis (white circle) and parapineal (yellow circle); scale bar = 10 µm. (E–F) Upper panel show a schematic of the SU5402 (or DMSO) treatment timeline (24–32 hpf) in control and *mib*-/- mutant embryos. Dot plots showing the number (E) and the mean position (F) of *gfi1ab* expressing parapineal (PP) cells at 48 hpf in control embryos treated with DMSO (light blue dots; n = 32) or with 5 µM SU5402 (dark blue dots, n = 31), and in *mib*-/- mutant embryos treated with DMSO (orange dots, n = 31) or with SU5402 (yellow dots, n = 41). The number of *gfi1ab*-positive cells is increased in *mib*-/- mutants embryos, regardless of whether they were treated with SU5402 or DMSO (DMSO control versus DMSO *mib*-/-, ** p-value=0.0042; SU5402 control versus SU5402 *mib*-/-, ** p-value=0.0086, Welch t-test). The parapineal fails to migrate in 52% of DMSO treated *mib*-/- mutant embryos (PP mean position between −15 µm to +15 µm, gray-shaded area), and this proportion decreases to 19% of SU5402 treated *mib*-/- mutant embryos; p-value=0.0139 on a Chi-square test and p=0.0103 on a Welch t-test on absolute value. (G) Upper panel shows a schematic of the SU5402 (or DMSO) treatment timeline (24–32 hpf) in control and *mib*-/- mutant embryos. Dot plot showing the number of *Tg(dusp6:d2EGFP)* expressing parapineal cells at 32 hpf in control embryos treated with DMSO (light blue; n = 20) or with

*Figure 4 continued on next page*

*Figure 4 continued*

5 μM SU5402 (dark blue, n = 18), and in *mib*-/- mutant embryos treated with DMSO (orange, n = 17) or with SU5402 (yellow, n = 21). The number of *Tg (dusp6:d2EGFP)+* cells is increased in DMSO treated *mib*-/- mutants versus controls (p-value=0.0077) but is decreased in SU5402 treated *mib*-/- mutants compared to DMSO treated *mib*-/- mutants (p-value=0.0248 in Welch's t-test). (H–I) Upper panel in H shows a schematic of heat shock timeline in *Tg (hsp70:ca-fgfr1)* embryos injected with *rbpja/b* morpholinos (MO). Dot plots showing the number (H) and the mean position (I) of *gfi1ab* expressing parapineal cells in control embryos that carry (purple dots; n = 55) or do not carry the *Tg(hsp70:ca-fgfr1)* transgene (light blue dots; n = 54), or in *rbpja/ b*MO injected embryos that carry (dark red dots; n = 73) or do not carry the *Tg(hsp70:ca-fgfr1)* transgene (orange dots; n = 67). In *rbpja/b* morphants expressing *Tg(hsp70:ca-fgfr1)*, the parapineal failed to migrate in 25% of embryos (n = 18/73; p-value=0.0007 Welch's t-test on absolute value and p-value=0.0232 in Chi-square test), a defect significantly higher than expected from merely adding the effects of activated receptor transgene and *rbpja/b* MO injections alone (p-value=0.0001 in Welch t-test on absolute value and p-value=0.0003 in Chi-square test). Data are representative of four (H–I), three (A–F) or two experiments (G). Source files used to generate dot plots and for statistical analysis are available in *Figure 4—source data 1*.

DOI: https://doi.org/10.7554/eLife.46275.016

The following source data is available for figure 4:

**Source data 1.** Source files for data used to generate dot plots in *Figure 4*.

DOI: https://doi.org/10.7554/eLife.46275.017

embryos injected with *rbpja/b MO* (4 to 8 ng *rbpja/b* MO); we chose this Notch context as parapineal migration defects are more modest than in *mib*-/- mutants. As previously described (*Roussigné et al., 2018*), widespread expression of constitutively activated receptor (CA-FgfR1) prevented parapineal migration in a small number of embryos (7% of embryos, n = 4/55, with a parapineal mean position between −15 and +15 μm; p-value=0.0011), while the parapineal consistently migrates in heat shocked control embryos not carrying the transgene (n = 54/54 with n = 51/54 leftwards and n = 3/54 toward the right) (*Figure 4I*). In the absence of the CA-FgfR transgene, *rbpja/b* morphant embryos displayed a migration defect at low frequency (7%; n = 5/67) (*Figure 4I*). However, in *rbpja/b* morphants expressing the activated receptor, the frequency of embryos in which the parapineal failed to migrate increased significantly (25% of embryos; n = 18/73) (*Figure 4I*, red versus orange and purple dots). Notch and FGF pathways appear to interact in this context as the increase in parapineal migration defects observed in *rbpja/b* morphant embryos expressing the activated Fgf receptor is significantly higher than expected from adding the effects of activated receptor transgene and *rbpja/b* MO injections alone (p-value=0.0001 in Welch t test on absolute value and p-value=0.0003 in Chi-square test). The increased frequency of parapineal migration defects occurs in the absence of significant changes in parapineal cell-type specification (*Figure 4H*); the mean number of *gfi1ab* expressing parapineal cells did not vary significantly between heat shocked *rbpja/b* morphant embryos with or without the CA-FgfR transgene (14 ± 6 versus 13 ± 5; p-value=0.39).

Taken together, our loss-of-function, gain-of-function and epistasis experiments argue that the Notch pathway is required for parapineal migration, and that it acts by restricting FGF activation in parapineal cells.

## Discussion

In this study, we address how activation of the FGF pathway is restricted within parapineal cells. We show that expression of the *Tg(dusp6:d2EGFP)* reporter transgene is broader in Notch loss-of-function contexts and is reduced following global activation of the Notch pathway. These changes in FGF activation correlate with defect in parapineal migration. Epistasis experiments show that decreasing or increasing FGF signaling can respectively rescue or aggravate parapineal migration defects in Notch loss-of-function contexts. We conclude that the Notch pathway participates in restricting the activation of FGF signaling to few cells of the parapineal cluster, thus promoting its migration.

### Notch effects on parapineal cell specification can be uncoupled from migration

Studies addressing the link between cell specification and migration are rare and, as such, our knowledge of whether and how these two processes are coordinated is limited. In the lateral line primordium (LLP) model, proper morphogenesis of the future neuromast at the trailing edge of the

primordium is required for migration (*Durdu et al., 2014*; *Kozlovskaja-Gumbrienė et al., 2017*; *Lecaudey et al., 2008*; *Nechiporuk and Raible, 2008*). In embryos treated with LY411575 from 22 to 32 hpf, the number of *gfi1ab* and *sox1a* expressing parapineal cells is strongly increased but neither parapineal migration nor *Tg(dusp6:d2EGFP)* expression are affected. Therefore, our data reveals that cell-type specification and migration can be uncoupled in the parapineal. Our results resemble previous observations describing uncoupling of specification and migration of cardiac cells during in heart development (*Davidson et al., 2005*).

Notch-mediated cell-cell communication is well described for its role on cell fate restriction and progenitors maintenance (*Cau and Blader, 2009*). In neural tissues or in the pancreas, for instance, inhibition of the Notch signaling pathway causes premature differentiation of the progenitor cells into mature differentiated cells (*Li et al., 2015*). In the lateral line system, Notch signaling is required to restrict sensory hair cell progenitor fate to a central cell in the forming pro-neuromast (*Matsuda and Chitnis, 2010*). Similarly, in the parapineal, the number of *gfi1ab* and *sox1a* expressing parapineal cells is affected in loss or gain of function for Notch. However, in both contexts, the parapineal rosette is formed and a normal number of *tbx2b* expressing parapineal cells is detected. Therefore, while in previous studies it was assumed that all described parapineal markers (*tbx2b*, *sox1a*, *gfi1*ab) label the same cells, our data indicate that the pool of *tbx2b*+ parapineal cells probably differs from a *sox1a/gfi1ab*+ pool. Our results also suggest that Notch signaling acts downstream of Tbx2b to control the transition from *tbx2b* expressing putative progenitors to differentiated parapineal cells expressing *sox1a/gfi1ab*.

## Notch acts upstream of FGF signaling to promote parapineal migration

In most models describing cross-talk between the Notch and FGF pathways, Notch signaling is described to act downstream of the FGF pathway. In the zebrafish LLP, for instance, ectopic activation of Notch signaling can rescue the formation of neuromast rosettes in absence of FGF pathway activity, indicating that Notch signaling is required downstream of Fgf signals to promote apical constriction and rosette morphogenesis (*Kozlovskaja-Gumbrienė et al., 2017*). Notch signaling is also described to be a downstream effector of the FGF pathway for epithelial proliferation in the pancreas in mammalian embryos (*Hart et al., 2003*). Our epistasis experiments suggest that the Notch pathway acts upstream of FGF signaling in parapineal cells to restrict FGF pathway activation and promote migration. Therefore, although in both the parapineal and the lateral line system, Notch signaling plays a similar role in restricting the number of cells with a particular fate (as mentioned above), the crosstalk with the FGF pathway appears to differ between the two models; Notch signaling acts upstream or downstream of FGF pathway to promote parapineal migration or neuromast morphogenesis, respectively.

## Notch signaling restricts activation of the FGF pathway to promote parapineal migration

Our data show that gain or loss of function for Notch respectively result in an increase or decrease in the number of *Tg(dusp6:d2EGFP)*+ cells in the parapineal, with both correlating with parapineal (PP) migration defects. The fact that the number of *Tg(dusp6:d2EGFP)*+ cells and not the mean d2GFP intensity of *Tg(dusp6:d2EGFP)*+ cells is affected in *mib*$^{-/-}$ mutants strongly suggests that it is the restriction of FGF activation to few parapineal cells that is important for correct migration rather than a specific absolute level of FGF activity in leading cells. These data, together with the fact that partial rescue of migration in SU5402-treated *mib*$^{-/-}$ mutants correlates with a decrease in the number of *Tg(dusp6:d2EGFP)*+ cells, indicate that the Notch pathway promotes migration by restricting the activation of FGF signaling to a few parapineal cells.

Notch signaling has previously been implicated in the migration of epithelial cells sheets (*Riahi et al., 2015*), trachea cells in *Drosophila* (*Ghabrial and Krasnow, 2006*; *Ikeya and Hayashi, 1999*) and in developing vertebrate blood vessels (*Siekmann and Lawson, 2007b*). In all these models of sheet or sprouting morphogenesis, migrating cells remains attached to the bulk of the tissue. The parapineal is thus the first described model of an isolated cluster of migrating cells in which Notch signaling modulates RTK signaling to define leading cells.

In both the tracheal and vascular systems, Notch-Delta signaling contributes to the selection of the tip cells by restricting the ability of follower cells to activate RTK signaling. The molecular

mechanisms underlying Notch mediated restriction of RTK signaling in these two models are not clear but could possibly involve the transcriptional control of RTK ligands or receptors (*Ghabrial and Krasnow, 2006*; *Ikeya and Hayashi, 1999*; *Siekmann and Lawson, 2007b*). As FgfR4, the only Fgf receptor found to be expressed in the parapineal, does not display restricted expression in the parapineal (*Regan et al., 2009*), it is unlikely that the Notch pathway acts through the control of this receptor.

How Notch signaling could modulate FGF pathway activity in parapineal cells remains an open question. The fact that some *rbpja/b* morphants display parapineal migration defects similar to *mib*$^{-/-}$ mutants strongly supports a role for the canonical Notch pathway rather than a Notch-independent role of the Mindbomb ubiquitin ligase in parapineal migration, although we cannot exclude that Mindbomb also regulates migration independently of the Notch pathway, for instance through modulation of Rac1 (*Mizoguchi et al., 2017*). As all parapineal cells are competent to migrate and to activate the FGF pathway (*Concha et al., 2003*; *Roussigné et al., 2018*), a parsimonious mechanism would be that lateral inhibition based cell-cell communication between parapineal cells within the rosette would modulate the capacity of a cell to activate/maintain or to inhibit FGF pathway activity. As described for the *C. elegans* vulva (*Berset et al., 2001*; *Yoo et al., 2004*), for instance, Notch signaling could directly promote the transcription of Ras/MAPK pathway inhibitors. Alternatively, Notch signaling could be required outside the parapineal to trigger mosaic FGF pathway activation in parapineal cells. Finally, there could be both parapineal-intrinsic and extrinsic requirements for Notch. In any case, as the ectopic expression of the constitutive activated FGF receptor (CA-FgfR1) does not completely block parapineal migration in a wild-type background and can rescue migration in *fgf8*$^{-/-}$ mutants (*Roussigné et al., 2018*), it appears that the restriction mechanism acts downstream of the receptor rather than at the level of receptor gene expression.

## When and where is Notch signaling required?

Given that LY411575 does not block migration in any of the tested time windows, we cannot be sure about the timing of the Notch requirement for migration *per se*. However, the fact that ectopic Notch signaling at late stages (26–28 hpf) can efficiently block migration argues for a role of Notch at the time when the parapineal initiates its migration and not earlier in development. While expression of the *Tg(Tp1:EGFP)* Notch reporter transgene is not robustly detected in parapineal cells, the fact that the expression of the Notch ligand *deltaB*, and occasionally *delta*A, can be detected in 1–2 parapineal cells after 28 hpf argues further for a role of Notch signaling within parapineal cells at the time migration starts.

If Notch signaling is indeed required at the time of migration initiation, it is unclear why LY411575 treatment (22 to 32 hpf) does not affect migration while it is able to trigger an increase in the number of *gfi1ab* and *sox1a*-positive parapineal cells. Classically, Notch signaling is thought to act through a lateral inhibition mechanism leading to a Notch-ON or Notch-OFF outcome between neighboring cells. Recent work, however, suggests that Notch acts in a level-dependent rather than an all-or-nothing manner (*Ninov et al., 2012*). In light of this, the differential effect of LY411575 on specification and migration of parapineal cells could reflect a different requirement in Notch signaling threshold, with a lower threshold of Notch signaling being required for migration and a higher one being required to control fate specification of parapineal cells. If LY411575 does not completely block Notch signaling, then residual Notch activity could be sufficient to promote the restriction of FGF activity and parapineal migration while not to limit the specification of *gfi1ab* and *sox1a*-positive cells. Consistent with the hypothesis that LY411575 treatment might be partially effective, we observed that the average number of *Tg(dusp6:d2EGFP)+* cells increases slightly in LY411575 treated embryos from 22 to 32 hpf, without reaching statistical significance. Given the high variability observed in the number of *Tg(dusp6:d2EGFP)+* cells in wild-type contexts, we might expect that parapineal migration is robust enough to tolerate a moderate increase in the number of *Tg(dusp6:d2EGFP)+* cells.

## Synergistic or parallel role of the Nodal and Notch pathways in restricting FGF pathway activation

The Notch signaling pathway has previously been implicated in regulating the expression of Nodal/TGF-β signal around the node and subsequently in the left lateral plate mesoderm (LPM)

(*Krebs et al., 2003*; *Raya et al., 2003*). In zebrafish, expression of a *nodal* related gene (*ndr3/south-paw*) in the left LPM is required for the later expression of a second *nodal* gene (*ndr2/cyclops*) in the left epithalamus, which is required for left-biasing parapineal migration (*Concha et al., 2000*; *Liang et al., 2000*; *Regan et al., 2009*). In *mib-/-* mutants, *rbpj* morphant embryos or in embryos treated with LY411575 from 8 to 22 hpf, the Nodal pathway is activated on both sides of the epithalamus. This requirement for Notch signaling in unilateral activation of the Nodal pathway in the epithalamus is consistent with an early role of Notch signaling in establishing the initial left right asymmetry and explains the partial randomization of parapineal migration we observe in contexts of early Notch loss-of-function.

Our previous results indicate that Nodal signaling contributes to restricting FGF pathway activation, as well as biasing it to the left (*Roussigné et al., 2018*). Here, we show that the restriction of FGF activity requires a functional Notch pathway. As mentioned above, the Nodal pathway is bilateral in the epithalamus of embryos with compromised Notch signaling. However, it is unlikely that the role of Notch in restricting FGF pathway activation depends on its ability to control Nodal signaling. Indeed, in other contexts of bilateral Nodal signaling, such as in embryos injected with morpholinos against *notail*, we have shown that FGF pathway activation ultimately becomes restricted to a few parapineal cells and parapineal eventually migrates (*Roussigné et al., 2018*); *Tg(dusp6:d2EGFP)* reporter expression in this context is no longer left lateralized, which correlates with the parapineal migrating either to the left or the right. In absence of Nodal signaling, as in *spaw* morphants, *Tg(dusp6:d2EGFP)* expression is generally less restricted within the parapineal and this correlates with delayed parapineal migration, indicating that the restriction of FGF activity is influenced by Nodal signaling as well as by the Notch pathway. How these two pathways interact to restrict the activation of FGF signaling is not known and future investigations will be needed to address whether Nodal and Notch pathway act in a synergistic way or in parallel to restrict the FGF pathway.

## Conclusion

Our study shows that Notch signaling is required for parapineal migration through its capacity to trigger cell state differences in FGF signaling and to restrict FGF activity to a few leading cells. As the function and cross-regulation of the FGF and Notch pathways might be conserved during the migration of invading cancerous cells, our results could help to understand these pathway interaction during metastasis.

# Materials and methods

### Key resources table

| Reagent type (species) or resource | Designation | Source or reference | Identifiers | Additional information |
|---|---|---|---|---|
| Genetic reagent (*D. rerio*) | *mib*[ta52b] | Itoh et al., 2003 | RRID:ZFIN_ZDB-GENO-071218-1 | |
| Genetic reagent (*D. rerio*) | *fgf8*[ti282a] (*ace*) | Reifers et al., 1998 | RRID:ZFIN_ZDB-GENO-980202-822 | |
| Genetic reagent (*D. rerio*) | *Tg(hsp70:Gal4)*[kca4] | Scheer et al., 2001 | RRID:ZFIN_ZDB-ALT-020918-6 | |
| Genetic reagent (*D. rerio*) | *Tg(UAS:myc-Notch1a-intra)*[kca3] | Scheer and Campos-Ortega, 1999 | RRID:ZFIN_ZDB-ALT-020918-8 | |
| Genetic reagent (*D. rerio*) | *Tg(hsp70:ca-fgfr1; cryaa:DsRed)*[pd3] | Marques et al., 2008 Neilson and Friesel, 1996 | RRID:ZFIN_ZDB-GENO-090127-1 | |
| Genetic reagent (*D. rerio*) | *Tg(dusp6:d2EGFP)*[pt6] | Molina et al., 2007 | RRID:ZFIN_ZDB-GENO-071017-5 | |
| Genetic reagent (*D. rerio*) | *Tg(Tp1bglob:EGFP)*[ia12] | Corallo et al., 2013 | RRID:ZFIN_ZDB-ALT-130115-3 | |
| Sequence-based reagent | *rbpja/su(H)1; rbpjb/su(H)2* | Echeverri and Oates, 2007 | ZFIN ID : ZDB-FISH-150901–13686 | |
| Antibody | Rabbit anti-GFP | Torrey Pines Biolabs | TP-401 RRID:AB_10013661 | 1/1000 |

*Continued on next page*

Continued

| Reagent type (species) or resource | Designation | Source or reference | Identifiers | Additional information |
|---|---|---|---|---|
| Antibody | Goat anti-rabbit IgG Alexa 488-conjugated | Molecular probe | A11034 | 1/1000 |
| Chemical compound, drug | LY411575 | MedChem Express | HY-50752 | *Rothenaigner et al., 2011* |
| Chemical compound, drug | SU5402 | Calbiochem | 572630 | *Mohammadi et al., 1997* |
| Other | Fast Red | Sigma Aldrich | F46-48 | |
| Other | ToPro-3 | Molecular probe | T3605 | 1/1000 |

## Fish lines

Embryos were raised and staged according to standard protocols (*Westerfield, 2000*). Embryos homozygous for *mindbomb* (*mib$^{ta52b}$*) (*Itoh et al., 2003*) and *fgf8* mutations (*fgf8 $^{ti282a}$/acerebellar/ ace; Reifers et al., 1998*) were obtained by inter-crossing heterozygous carriers. Carriers of the *fgf8$^{ti282a}$* allele were identified by PCR as described previously (*Roussigné et al., 2018*). *mib$^{ta52b+/-}$* carriers were identified by PCR genotyping using primers 5'-GGTGTGTCTGGATCGTCTGAAGAAC-3' and 5'-GATGGATGTGGTAACACTGATGACTC-3' followed by enzymatic digestion with NlaIII. *Tg (hsp70:Gal4)$^{kca4}$* and *Tg(UAS:myc-Notch1a-intra)$^{kca3}$* transgenic lines have been described previously (*Scheer et al., 2001; Scheer and Campos-Ortega, 1999*) and identified by PCR genotyping following ZIRC genotyping protocols. Embryos carrying the *Tg(hsp70:ca-fgfr1; cryaa:DsRed)$^{pd3}$* transgene (*Marques et al., 2008; Neilson and Friesel, 1996*) were identified by the presence of DsRed expression in the lens from 48 hpf or, by PCR at earlier stages as described previously (*Gonzalez-Quevedo et al., 2010*). *Tg(dusp6:d2EGFP)$^{pt6}$* (*Molina et al., 2007*) and *Tg(Tp1bglob:EGFP)$^{ia12}$* (*Corallo et al., 2013; Parsons et al., 2009*) lines were used as reporters for the FGF and Notch pathways, respectively. Embryos were fixed overnight at 4°C in BT-FIX (*Westerfield, 2000*), after which they were dehydrated through an ethanol series and stored at −20°C until use.

## Ethics statement

Fish were handled in a facility certified by the French Ministry of Agriculture (approval number A3155510). The project has received an agreement number APAFIS#3653–2016011512005922. Anaesthesia and euthanasia procedures were performed in Tricaine Methanesulfonate (MS222) solutions as recommended for zebrafish (0.16 mg/ml for anaesthesia, 0.30 mg/ml for euthanasia). All efforts were made to minimize the number of animals used and their suffering, in accordance with the guidelines from the European directive on the protection of animals used for scientific purposes (2010/63/UE) and the guiding principles from the French Decret 2013–118.

## LY411575 treatment

Embryos collected from *Tg(dusp6:d2GFP)$^{pt6}$* carriers outcrossed with wild-type fish or with heterozygous *mib$^{ta52b}$* were dechorionated and treated from 8 to 22 hpf or 22 to 32 hpf with 30 µM, 100 µM or 200 µM of LY411575 (MedChem; *Rothenaigner et al., 2011*); control embryos were treated with an equal volume of DMSO diluted in E3 medium. *Tg(dusp6:d2GFP)* expressing embryos were incubated in LY411575 at 28°C and fixed at 32 hpf for immune-staining against EGFP; sibling embryos, not carrying the *Tg(dusp6:d2GFP)* transgene, were fixed at indicated time (32 hpf, 36 hpf or 48 hpf) for *in situ* hybridization against different parapineal markers.

## SU5402 drug treatment

Embryos collected from in-crosses between *mib$^{ta52b}$* and *Tg(dusp6:d2EGFP)*; *mib$^{ta52b}$* mutants were dechorionated and treated with 5 µM SU5402 (Calbiochem; *Mohammadi et al., 1997*) by diluting a 10 mM DMSO based stock solution in E3 medium; control embryos were treated with an equal volume of DMSO diluted in E3 medium. All embryos were incubated in SU5402 at 28°C from 24 hpf to 32 hpf. *Tg(dusp6:d2EGFP)* embryos were fixed at 32 hpf to analyze the d2EGFP expression pattern in both *mib$^{ta52b}$* mutants and sibling embryos; the remaining half of embryos were fixed at 48 hpf to analyze parapineal migration.

## Morpholino injections

Morpholino oligonucleotides targeting both *rbpja/su(H)one* and *rbpjb/su(H)2* (*Echeverri and Oates, 2007*) were dissolved in water at 3 mM. The resulting stock solution was diluted to working concentrations (0.3 mM, 2.5 ng/nl) in water and Phenol Red before injection of 1.5 nl (about 4 ng) or 3 nl (8 ng) into embryos at the one-cell stage. Embryos were subsequently fixed and processed for *In situ* hybridizations or antibody labeling.

## Heat shock procedure

Ectopic expression of the intracellular domain of Notch receptor (NICD) was induced in *Tg(hsp70:gal4); Tg(UAS:myc-Notch1a-intra)* double transgenic embryos by incubating them at 39°C for 45 min starting at different time points (22 hpf, 24 hpf, 26 hpf, 28 hpf or 32 hpf). Embryos were then incubated at 28.5°C and fixed at 36 hpf to analyze *Tg(dusp6:d2EGFP)* expression or at 48 hpf for *in situ* hybridizations against indicated parapineal markers. Ectopic expression of CA-FgfR1 was induced in *Tg(hsp70:ca-FgfR1; cryaa:DsRed)^{pd3}* heterozygote embryos by performing a first heat shock at 25–26 hpf (39°C, 45 min) and a second short heat shock (39°C, 15 min) 3 hours later (28–29 hpf) in order to cover the entire period of parapineal migration. Control embryos are non-transgenic, or carry only *Tg(UAS:myc-Notch1a-intra)* or *Tg(hsp70:Gal4)* transgene, and were heat-shocked under the same conditions as the *Tg(hsp70:ca-FgfR1)* transgenic or *Tg(hsp70:gal4); Tg(UAS:myc-Notch1a-intra)* double transgenic embryos.

## *In situ* hybridization and immunostaining

*In situ* hybridizations were performed as described previously (*Roussigné et al., 2018*), using antisense DIG labeled probes for *gfi1ab* (*Dufourcq et al., 2004*), *sox1a* (*Clanton et al., 2013*), *pitx2c* (*Essner et al., 2000*), *tbx2b* (*Snelson et al., 2008*), *deltaA* and *deltaB* (*Haddon et al., 1998*). *In situ* hybridizations were completed using Fast Red (from Sigma Aldrich) as an alkaline phosphatase substrate. Immunostainings were performed in PBS containing 0.5% triton using anti-GFP (1/1000, Torrey Pines Biolabs) and Alexa 488-conjugated goat anti-rabbit IgG (1/1000, Molecular Probes). For nuclear staining, embryos were incubated in ToPro-3 (1/1000, Molecular Probes) for 1 hr as previously described (*Roussigné et al., 2018*).

## Image acquisition

Confocal images of fixed embryos were acquired on an upright Leica SP8 confocal microscope using the resonant fast mode and either an oil x63 (aperture 1.4) or x20 (aperture 1.4) objective. Confocal stacks were analyzed using ImageJ software. Figures were prepared using Adobe Photoshop software.

## Quantification of the number and position of *Tg(dusp6:d2EGFP)* positive parapineal cells

The position and number of parapineal cells positive for the *Tg(dusp6:d2GFP)* transgene were analyzed using ROI Manager tool on ImageJ software as previously described (*Roussigné et al., 2018*). The mean intensity of the d2EGFP staining was quantified in an area corresponding to the cell nucleus and the same intensity threshold was used in the different experimental contexts to determine if a cell was *Tg(dusp6:d2EGFP)* positive or not. The total number of parapineal cells was estimated at 32 hpf by counting nuclei in the parapineal rosettes using Topro-3 nuclear staining.

## Quantification of the number and position of *gfi1ab*, *sox1a*, *tbx2b* positive parapineal cells

The position and number of *gfi1ab*, *sox1a* or *tbx2b* positive parapineal cells were analyzed using the Multipoint tool on ImageJ software and determined as the centre of the cell nucleus detected with the Topro-3 nuclear staining as previously described (*Roussigné et al., 2018*). The position of each parapineal cell was measured relative to the brain midline (reference origin = 0) as determined by a line passing along the lumen of the epiphysis. For each embryo, we calculated the number of labeled parapineal cells and their mean position. To avoid any bias, data in *Figure 4E–F* were analyzed blind for DMSO versus SU5402 treatment and data describing LY411575 treatment (*Figure 2I–N*,

*Figure 2—figure supplement 1*) were analyzed blind for *mib* genotypes (*mib* ± heterozygotes versus +/+ controls).

## Statistical analysis

Statistical comparisons of datasets were performed using R Studio or GraphPad Prism software. For each dataset, we tested the assumption of normality with Shapiro-Wilks tests and variance homogeneity with F tests. As datasets on the number of parapineal cells were usually normal and often of unequal variances, they were compared using unpaired Welch t-tests; means (± SEM) are indicated as horizontal bars on dot plots. Data on parapineal mean position usually did not distribute normally (because of few embryos with the parapineal on the right) and were compared using the Wilcoxon rank sum non-parametric tests; we also compared parapineal mean position datasets with Welch t-tests using absolute values to discriminate between a defect in laterality (i.e. left-right randomization of parapineal migration) and a defect in migration *per se* (i.e. distance from the midline only). Most data are representative of at least two and more often three independent experiments and the number of biological replicates is mentioned in the figure legends for each graph. Means (± SD) are indicated in the text.

## Acknowledgements

We are grateful to all members of the Blader lab, especially Elise Cau and Julie Batut for critical reading of the manuscript and Aurélie Quillien for insightful discussions. We also thank Eric Theveneau and Matthias Carl for critical reading of the manuscript, as well as Xiaobo Wang and Steve W Wilson for their input. We thank Brice Ronsin from the Toulouse RIO Imaging platform and Aurore Laire for fish care. This work was supported by the Centre National de la Recherche Scientifique (CNRS), the Institut National de la Santé et de la Recherche Médicale (INSERM), Université de Toulouse III (UPS), the Fondation pour la Recherche Médicale (FRM; DEQ20131029166), the Fondation ARC (PJA 20131200173), the Agence nationale de la recherche (ANR-16-CE13-0013-01) and the China Scholarship Council (CSC No.201504910807) for PhD funding for Lu Wei.

## Additional information

### Funding

| Funder | Grant reference number | Author |
|---|---|---|
| Fondation pour la Recherche Médicale | DEQ20131029166 | Patrick Blader |
| Fondation ARC pour la Recherche sur le Cancer | PJA 20131200173 | Patrick Blader |
| Agence Nationale de la Recherche | ANR-16-CE13-0013-01 | Patrick Blader |
| China Scholarship Council | CSC No.201504910807 | Lu Wei |
| Centre National de la Recherche Scientifique | | Myriam Roussigné |
| Université Toulouse III - Paul Sabatier | | Myriam Roussigné |
| Inserm | | Patrick Blader |

The funders had no role in study design, data collection and interpretation, or the decision to submit the work for publication.

### Author contributions

Lu Wei, Data curation, Formal analysis, Validation, Investigation, Visualization, Methodology, Writing—original draft; Amir Al Oustah, Validation, Investigation; Patrick Blader, Supervision, Funding acquisition, Writing—review and editing; Myriam Roussigné, Conceptualization, Data

curation, Formal analysis, Supervision, Funding acquisition, Validation, Investigation, Visualization, Methodology, Writing—original draft, Project administration, Writing—review and editing

### Author ORCIDs
Patrick Blader  http://orcid.org/0000-0003-3299-6108
Myriam Roussigné  https://orcid.org/0000-0002-4240-4105

### Ethics
Animal experimentation: Fish were handled in a facility certified by the French Ministry of Agriculture (approval number A3155510). The project has received an agreement number APAFIS#3653-2016011512005922. Anaesthesia and euthanasia procedures were performed in Tricaine Methane-sulfonate (MS222) solutions as recommended for zebrafish (0,16mg/ml for anaesthesia, 0,30 mg/ml for euthanasia). All efforts were made to minimize the number of animals used and their suffering, in accordance with the guidelines from the European directive on the protection of animals used for scientific purposes (2010/63/UE) and the guiding principles from the French Decret 2013-118.

### Decision letter and Author response
Decision letter https://doi.org/10.7554/eLife.46275.020
Author response https://doi.org/10.7554/eLife.46275.021

## Additional files

### Supplementary files
• Transparent reporting form
DOI: https://doi.org/10.7554/eLife.46275.018

### Data availability
All informations on datasets are provided in the eLife transparent reporting form and all data generated or analysed during this study are included in the manuscript and supporting files. Source data files required to reproduce dot plots and statistical analysis have been provided as an excel file for all main Figures 1-4.

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
