## [Decision Letter]

[Editors’ note: the authors were asked to provide a plan for revisions before the editors issued a final decision. What follows is the editors’ letter requesting such plan.]

Thank you for sending your article entitled "Notch signaling restricts FGF pathway activation in parapineal cells to promote their collective migration" for peer review at *eLife*. Your article is being evaluated by Marianne Bronner as the Senior Editor, a Reviewing Editor, and three reviewers.

Given the list of essential revisions, including new experiments, the editors and reviewers invite you to respond within the next two weeks with an action plan and timetable for the completion of the additional work. We plan to share your responses with the reviewers and then issue a binding recommendation.

In particular, the reviewers have several major concerns:

1) The reviewers wondered whether it was the levels of FGF activity rather than restricted FGF activity that may be playing an important role. To test this, they could us additional markers as a readout for FGF activity.

2) A crucial experiment is to determine in which cells Notch signaling is active. Without having this information, one cannot formulate a hypothesis of how Notch might be regulating FGF signaling.

3) While the study provides insights about when Notch activity is required, much less can be said about exactly where it is required. The authors can speculate in the discussion but must remain circumspect about this issue.

4) Although it would be interesting to know which Notch receptors and ligands are at play, this seems beyond the scope of what is necessary for this study.

Below are the full reviews from the three reviewers which we hope you find helpful.

*Reviewer #1:*

The manuscript describes how Notch signaling affects parapineal cell migration and cell type specification in the zebrafish brain. The manuscript is clearly written and illustrated, and the Discussion section is informative and thoughtful. The authors demonstrate that loss of Notch signaling leads to an expansion of the FGF pathway and defects in parapineal migration and laterality, and an increase in the number of *gfi1ab* and *sox1a* expressing parapineal cells. They further demonstrate that the roles of Notch signaling in cell type specification, cell migration and laterality of cell migration are uncoupled. Only the migration and laterality defects are FGF dependent, whereas the effect of Notch signaling on cell type specification is not. The authors also show that these different functions of Notch signaling occur at different time points of development. Cell migration to the left side (laterality) depends on Notch signaling before migration starts (8-22 hpf) by restricting Nodal signaling to the left side of the epithalamus. Whereas cell type specification occurs later. The main conclusion of the manuscript is that proper parapineal cell migration depends on Notch-dependent restriction of FGF signaling to just a few tip cells. The mechanisms by which Notch restrict FGF signaling is unknown.

The results are clearly illustrated and the majority of the data interpretation are sound. However, the manuscript would profit from a more detailed description of the different cell types. Also, the fact that suboptimal doses of SU5402 rescue the *mib* migration phenotype should be discussed/analyzed in more detail, as the results might suggest that it is not important that only the tip cells express FGF signaling but rather that these tip cells express the correct level of FGF signaling. Also, there is no description of where Notch signaling is active. As Notch reporter lines exist, the authors should add this information and discuss it.

Specific comments:

1) Subsection “The parapineal of *mindbomb* mutant embryos display expanded FGF pathway activation”: The number of parapineal cells is slightly increased but the number of *sox1a* positive cells is not increased (which is a marker for parapineal cells). Are there sox1a-negative parapineal cells or is the difference between Figure 1F and 1G due to biological noise?

2) Figure 1A: please indicate in the figure legend or figure what structures are outlined by yellow and white dotted lines.

3) Subsection “Loss of Notch signaling results in an increase in the number of certain parapineal cell subtypes”: A more detailed characterization of *tbx2b, sox1a* and *gfi1a* expressing cells is necessary. Are *sox1a* and *gfi1ab* expressed in the same cells?

4) Subsection “Loss of Notch signaling results in an increase in the number of certain parapineal cell subtypes” (and subsection “Notch effects on parapineal cell specification can be uncoupled from migration”) the authors conclude that Notch probably controls the transition from *tbx2b*-expressing putative progenitors to *sox1a/gfi1ab*-expressing differentiated cells. After loss of Notch signaling in *mib* mutants, the number of *sox1a/gfi1ab* positive cells increases, whereas the *tbx2b* positive cell number remains the same. How do the authors envision that the progenitor pool stays the same? Normally, defects in Notch dependent lateral inhibition should lead to the increase of one cell population at the expense of another. Do the progenitor cells divide asymmetrically and only the future *sox1a/gfi1ab* cells proliferate more? In this scenario *tbx2b*-expressing progenitor cells are independent of Notch signaling. In many other systems progenitor maintenance crucially depends on Notch signaling. Either the parapineal is different or the *tbx2b* positive cells are not self-renewing progenitors? Please discuss in more detail what is known about *tbx2b, sox1a* and *gfi1ab* positive cells.

5) Regarding the hs:NICD experiments: Are control embryos also heat shocked? Please label figure accordingly.

6) Figure 2I: The time line of treatment shown above the graph is very useful. It would facilitate the interpretation even further if it was indicated at what time point (32 hpf) migration begins.

7) The fact that ectopic Notch signaling at late stages (26-28 hpf) can efficiently block migration argues for a role of Notch at the time when the parapineal initiates its migration. If Notch signaling is indeed required just before migration, it is unclear why LY411575 treatment (22 to 32 hpf) does not affect migration. Maybe activation of Notch by inducing NICD until 28hpf persists much longer and NICD is still active after 28hpf. Therefore, embryos should be treated with LY411575 beyond 32hpf, as Notch might be required for migration during these later stages.

8) Subsection “Notch activity is required early for unilateral activation of the Nodal pathway in the epithalamus”: 'To determine whether the parapineal migration defects we observed (add: in *mib* mutants) might be a direct…. In sentence in the paragraph above the authors conclude that there was no migration defect in LY411575 treated embryos, which is confusing.

9) Subsection “Decreasing FGF signaling rescues the parapineal migration defects in loss of Notch context while increasing FGF signaling aggravates it”. The authors show that suboptimal doses of the FGF inhibitor SU5402 rescue to some degree the migration defect in *mib* mutants, which are characterized by FGF signaling activation in all migrating cells. This experiment actually argues that it is not important that the tip cells possess more FGF signaling then the follower cells but rather that FGF signaling needs to be expressed at the correct level. The authors should discuss the results of this experiment in more detail.

10) Subsection “Decreasing FGF signaling rescues the parapineal migration defects in loss of Notch context while increasing FGF signaling aggravates it”: 'Taken together, our.…experiments argue that the Notch pathway is required for parapineal migration, and that it acts by restricting FGF activation in parapineal cells.' However, in subsection “Notch activity is required early for unilateral activation of the Nodal pathway in the epithalamus” the authors argue that Notch and FGF act synergistically during migration, as *rbpj* morphant:cafgfr1a embryos display a more severe migration phenotype then CAfgfr1 embryos alone. Therefore, one has to conclude that Notch and FGF partially act in parallel on parapineal migration, correct?

11) Epistasis: Notch and FGF could also act in a feedback loop. Is Notch signaling affected after manipulation of FGF signaling?

Reviewer #2:

The paper by Lu et al., is an excellent follow up on previous work from Roussigne which demonstrated a role for restricted FGF signaling in collective migration of the parapineal cells. In this study the authors make good use of a combination of transgenic tools, mutants and chemical inhibitors to demonstrate the role of Notch signaling in a particular time window in restricting FGF signaling to a limited number of parapineal cells to determine asymmetric collective migration. They show that it is required after a relatively early stage when Notch is required to determine asymmetric Nodal signaling and after a stage is required to restrict the number of cells with *gf1ab* and *sox1a* positive cells. The experiments are presented in a logical order, the data is presented well, and its analysis supports the conclusions of the authors.

I have no major problems with the paper in its current form. My issues were only tangentially related to the study.

At various points in the Introduction and Discussion section the authors make comparisons with the role of FGF and Notch signaling in the lateral line system. I found those references confusing and potentially misleading. I would have liked to hear more about the role of Notch signaling in restricting migratory potential in tracheal cells and endothelial cells in vascular development. While the tracheal and vascular systems are referenced, the comparison could be elaborated on.

In the Introduction the authors suggest that the Dalle Nogare paper shows a role for FGF signaling in maintaining cluster cohesion. It is not clear what the authors have in mind here because the primary role of FGF shown in that paper is to suggest a role for FGFs released by leading cells in providing a guidance cue for collective migration of trailing cells. Their description of FGFs role in cluster cohesion is not essential for any point they wish to establish in this study and may only serve to establish a misleading impression about conclusions in the Dalle Nogare paper in the minds of readers for those not familiar with that paper.

Similarly, the authors contrast the role of Notch, downstream of FGF, in determining apical constriction in forming neuromasts in the migrating primordium, with the role of Notch in restricting the number of cells with FGF activity in the context of parapineal migration. The authors are not wrong here but they may give readers that not familiar with the Lateral Line system a simplistic idea about the role of Notch signaling in the primordium, where, as in the parapineal, it has multiple sequential roles. For example, first in restricting sensory hair cell progenitor fate to a central cell, and subsequently in determining Notch activation in neighboring cells to consolidate morphogenesis of epithelial rosettes. It is true that activation of Notch promotes apical constriction in the absence of FGF signaling and therefore in this context, downstream of FGF signaling and different in its epistatic relationship from the role of Notch in the parapineal. However, in the context of lateral inhibition and in restricting the number of cells with a particular fate/activity there may be important similarities in the role of Notch in the lateral line primordium and in parapineal cells. Without such a clarification, I felt the contrasting of the role of Notch in the parapineal and primordium could be potentially misleading.

A question that remained unanswered, which I was curious about and is not necessary in my mind for acceptance of the paper- which cells does Notch activity required in for its role in restricting migratory potential, which Notch ligands and receptors are required for this process and does their spatial distribution or that of downstream target genes provide insight about where Notch activity if required for restriction of FGF signaling in the parapineal.

Reviewer #3:

In this study by Wei et al., the authors investigate collective cell migration of parapineal cells during development using zebrafish as a model system. The authors found that global inhibition of Notch signaling, either by a chemical inhibitor or a MO-mediated knockdown, increased the number of cells that activate FGF signaling. As FGF is required for the parapineal migration and it is typically restricted to a few leading cells, this blocked the parapineal migration. Conversely, upregulating Notch signaling led to the loss of FGF signaling and also defects in its migration. These manipulations also affected a number of parapineal cells. Finally, the authors modulated FGF signaling in both Notch loss- and gain-of-function conditions. Decreasing FGF under this condition resulted in a partial rescue of the migration, whereas gain of FGF signaling worsened the migration phenotype. Interestingly, these manipulations did not affect the number of parapineal cells, indicating that these two roles of Notch signaling are distinct. Based on these data, the authors concluded that Notch restricts FGF signaling to a few leading parapineal cells, a process required for the proper migration. In addition, during later stages, Notch signaling controls the number of parapineal cells and this particular role of Notch can be uncoupled from its earlier role in restricting FGF signaling.

Overall, the conclusions are based on carefully done and controlled experiments (with a few exceptions). And while they provide some interesting insights into mechanisms of parapineal migration, the study is quite descriptive and does not go beyond the superficial analysis of cellular phenotypes. It is also not clear which cells express Notch ligands and receptors and whether effects on FGF signaling are direct. In summary, the overall findings are quite specialized and somewhat incremental and, thus, more suited for a specialized audience.

Major Comments:

Subsection “Loss of Notch signaling results in defects in parapineal migration”: Indicates that migration of the parapineal failed in 1/3 of *mib-/-*. I don't believe the authors discussed why this is the case/why the phenotype is partially penetrant in *mib* mutants? As well as why this phenotype was partially penetrant in the *rbpj* MO-injected embryos (with only 13% failing to migrate).

*rbpj a/b* MO: given specific guidelines for MOs (Stainier et al., 2017), these experiments do not seem to be appropriately controlled.

The results related to the inhibitor treatment seem odd. If the γ-secretase inhibitor, LY411575, is specific and Notch is playing a role in migration and specification of this system, I would expect to see a result with treatment from the 22-32 hpf time period. This drug and it has only been published in two zebrafish papers. Why not use the γ-secretase inhibitor DAPT that has been utilized in many other studies. Related to this, I am confused that 8-22 hour inhibitor treatment did not interfere with the *Tg(dusp6:d2EGFP)* transgene expression, neither did the 22-32 hour treatment. What is then the time window for Notch signaling that controls *Tg(dusp6:d2EGFP)* expression?

Throughout the manuscript, it is not clear where Notch signaling is active/ where Notch signaling components are expressed. It would be helpful to see the expression of where particular Notch signaling components are expressed during the time points of interest to further validate the reason behind studying Notch signaling. In other words, is Notch acting directly on parapineal cells?

[Editors’ note: formal revisions were requested, following approval of the authors’ plan of action.]

Further to my previous email, the editors and reviewers have considered your plan and invite you to proceed with your revisions as proposed. The asked us to pass on the following comments:

The authors should expand their discussion, add the discussion points raised by the reviewers and add relevant figure panels.

Specific comments:

In Figure 3 for reviewers they should add panels that only show the red signal.

As LY411575 does not inhibit migration at any time point, the authors should test if this drug is able to completely reduce Notch signaling. Is the signal in the TP Notch reporter line gone after treatment? Possibly, a small amount of Notch signaling is sufficient to allow for migration.

The expression data of Notch pathway members should be included and discussed in the manuscript.

The text describing the different cell types should be clarified, as it is currently unclear which genes mark possibly the same cells or which genes label all cells.

[Editors' note: further revisions were requested prior to acceptance, as described below.]

Thank you for submitting your article "Notch signaling restricts FGF pathway activation in parapineal cells to promote their collective migration" for consideration by *eLife*. Your article has been reviewed by Marianne Bronner as the Senior Editor, a Reviewing Editor, and two reviewers. The reviewers have opted to remain anonymous.

The reviewers have discussed the reviews with one another and the Reviewing Editor has drafted this decision to help you prepare a revised submission. Please aim to submit the revised version within two months.

The reviewers felt that manuscript is much improved and addressed the main concerns to: (1) provide evidence that some Notch pathway signaling members are expressed in the parapineal during migration; (2) further investigate effects of the γ-secretase inhibitor treatment on the Notch pathway activation, and (3) clarify the dynamics of parapineal marker gene expression during PP development. However, the evidence that Notch pathway acts within parapineal cells remains weak: deltaB and the reporter expression are not robust enough to make that conclusion, and functional experiments testing this hypothesis are absent. Therefore, please tone down that conclusions and present alternative possibilities.

We will look forward to hearing from you with a revised article with tracked changes, and a response letter (uploaded as an editable file) describing the changes made in response to the decision and review comments.

---

## [Author Response]

[Editors’ note: what follows is the authors’ plan to address the revisions.]

Given the list of essential revisions, including new experiments, the editors and reviewers invite you to respond within the next two weeks with an action plan and timetable for the completion of the additional work. We plan to share your responses with the reviewers and then issue a binding recommendation.In particular, the reviewers have several major concerns:1) The reviewers wondered whether it was the levels of FGF activity rather than restricted FGF activity that may be playing an important role. To test this, they could us additional markers as a readout for FGF activity.

Our data show that gain or loss of function for Notch results respectively in an increase or decrease in the number of *Tg(dusp6:d2EGFP)*+ cells in the parapineal, with both resulting in parapineal (PP) migration defects. These data, together with the fact that partial rescue of migration (Figure 4F) correlates with a decrease in the number of *Tg(dusp6:d2EGFP)*+ cells (Figure 4G), led us to conclude that FGF signaling needs to be restricted within PP cells. Reviewer 1 proposed that a correct level of FGF signaling in the leading cell might be required rather than or in addition to a restriction of the pathway activation to a few cells. This alternative interpretation is indeed a possibility. As explained below, our capacity to analyze the absolute level of FGF signaling is limited by the complexity of the FGF pathway and the tools at our disposal. However, here we provide a fuller set of arguments in favor of a requirement for the restriction in activation over a specific level of FGF activity being necessary for correct parapineal migration.

We analyzed the expression of additional markers as readout for FGF activity prior to our original submission. The activation of the FGF pathway leads to the transcription of various target genes that either encode positive effectors, such as members of the PEA3 subfamily of ETS domain transcription factor (Etv5, ERM and Pea3), or feedback inhibitors (Sprouty 1,2,4 and sef) of the FGF pathway (Tsang and Dawid, 2004). As part of the so-called *fgf8* syn-expression group, these genes have been described in the literature or the ZFIN database to be expressed in the diencephalon in a pattern similar to Fgf8 and Fgf3, but their detailed expression pattern in the epithalamus had not been addressed. We analyzed the expression pattern of these genes in the epithalamus at cellular resolution and found that *erm* and *pea3* are expressed broadly in the epithalamus, in both the parapineal and presumptive habenulae, while *etv5* is more specifically detected in the PP and enriched at the migration front (see Author response image 1 for reviewers). We also found that the *Tg(sprouty1:YFP)* transgene is expressed in some PP cells, although we could not detect endogenous *sprouty1* expression; the expression of *sprouty 2,4* or *sef* is only detected in the habenulae but not in the PP.

**Author response image 1. respfig1:** Expression of others FGF target genes in the epithalamus. Confocal sections of the head of *Tg(dusp6:d2EGFP)* embryos (A-C’’’) at 36 hpf after a whole-mount *in situ* hybridization against *erm* (red, A-A’’’), *pea3* (red, B-B’’’), *etv5a* (red, C-C’’’) or of Tg(*sprouty1:YFP)* embryos at 30 hpf (D-D’) or 36 hpf (E, E’) after immunostaining against GFP (Green; A-E’); the pictures are merged with cell nuclei staining (Topro-3 in grey) that makes brain structures visible and specially the parapineal (yellow circle). A’’-C’’’, D’-E’ show Zoom in of merge pictures. Embryos are viewed dorsally with anterior up.

These data show that FGF target genes other than *dusp6* are focally expressed in the parapineal (*etv5* and perhaps *sprouty1*) and enriched at the migration front. Nonetheless, their expression does not systematically overlap with *Tg(dusp6:d2EGFP)* reporter transgene expression, highlighting the complexity of the FGF signaling pathway: although the classic FGF target genes are all expressed in the same territories as Fgf8 (the epithalamus), their expression pattern are somewhat distinct at a cellular level. As such, we do not think that performing *in situ* against these other markers will allow us to quantify absolute level of FGF activity better than we have already managed in the *Tg(dusp6:d2EGFP)* line, and thus to fully address the alternative hypothesis raised by reviewer 1.

The complexity revealed by the expression pattern of other FGF target genes led us to choose to focus on *Tg(dusp6:d2EGFP)* transgene expression only as we have previously shown that its expression reflects an Fgf8 dependent activation of one branch of the FGF pathway that is required for parapineal migration (Roussigné et al., 2018). To avoid any bias in counting the number of *Tg(dusp6:d2EGFP)+* cells, in our study, we quantified the mean intensity of d2EGFP expression for each PP cell and used, for each experiment, the same intensity threshold to determine if a cell was *Tg(dusp6:d2EGFP)* positive or not in the various contexts (see Material and Methods). While not originally designed to address the level of FGF pathway activation *per se*, these data can be used to analyze the mean intensity of *Tg(dusp6:d2EGFP)*+ cells in addition to the number of *Tg(dusp6:d2EGFP)*. As such, we revisited the data concerning *mib-/-* mutants versus wildtype (Figure 1) or in SU5402 versus DMSO treated wild-type and *mib-/-* (Figure 4) relative to the general level of d2GFP expression. Treatment with the suboptimal dose of SU5402 (5µM) in *mib-/-* context triggers a partial rescue of PP migration together with a decrease in both the number (Figure 4G) and the mean intensity of *Tg(dusp6:d2EGFP)*+ cells (Author response image 2): although the p-value is higher than the 0,05 significance threshold (p=0,59), we observe a clear tendency for a decrease in the mean intensity of *Tg(dusp6:d2EGFP)*+ cells upon SU5402 treatment (See Author response image 2). Therefore, in this experiment, we cannot conclude if the SU5402 mediated partial rescue is due to an overall decrease in the number or in the intensity of *Tg(dusp6:d2EGFP)* FGF reporter. However, when we compare the mean intensity of *Tg(dusp6:d2EGFP)*+ cells in *mib-/-* versus wild-type embryos, both in data from the original Figure 4 and Figure 1, we clearly observe that the number of *Tg(dusp6:d2EGFP)*+ increases (See Author response image 2 for Reviewers) while the mean intensity does not change significantly in *mib-/-* mutants (See Author response image 2 for Reviewers). Therefore, PP migration defects in *mib-/-* mutants correlate with an increase in the number rather than the mean d2GFP intensity of *Tg(dusp6:d2EGFP)*+ cells. This strongly suggests that, rather than the level of FGF activity, it is the restriction of FGF activation to few parapineal cells that is important for correct migration.

**Author response image 2. respfig2:** The number but not the mean intensity of *Tg(dusp6:d2EGFP*) expressing cells is increased in *mib-/-* mutants. (**A-B**) Dot plots showing the mean GFP fluorescence intensity (A, B) of *Tg(dusp6:d2EGFP)* expressing parapineal cells in the first (**A**) and second experiment (**B**) in controls (blue dots; A, n=21; B, n=15) or in *mib^-/-^*mutant embryos (orange dots; A, n=21; B, n=13) at 32 hpf stage. (**C-D**); we chose to present data for the 2 experiments separately as the mean intensity threshold chosen to define *Tg(dusp6:d2EGFP)+* cells differ between both experiments. (**C-D**) Dot plots showing the number of *Tg(dusp6:d2EGFP)* expressing parapineal cells in the first (**C**) and second experiment (**D**); merged data for both experiments are presented in the Manuscript as Figure 1E. (E-F) Dot plots showing the mean fluorescence intensity of Tg(dusp6:d2EGFP) expressing parapineal in control embryos treated with DMSO(light blue; A, n=20) or with 5 µM SU5402 (dark blue; A, n=18), and in mib-/-mutant embryos treated with DMSO (orange; A, n=17) or with SU5402 (yellow; A, n=21) at 32 hpf stage. Mean ± SEM is indicated; ns, p-value>0,05; * p-value<0,05; ** p-value<0,01 in Welch t-test.

We would be happy to mention this ‘level of FGF’ alternative interpretation in the discussion of a revised manuscript and present the d2GFP intensity data in a Supplement Figure to support our interpretation. Alternatively, we could include these data as new panels in Figure 1 and Figure 4 if the reviewers think it is appropriate.

2) A crucial experiment is to determine in which cells Notch signaling is active. Without having this information, one cannot formulate a hypothesis of how Notch might be regulating FGF signaling.

To address whether the Notch pathway is active in the parapineal, we checked the expression of a well-characterized Notch reporter transgene, Tg(TP1:EGFP) (Parson et al., 2009; Corallo et al., 2013), at 30 and 36 hpf. We found that, despite being expressed at very high level in the epiphysis, this transgene was not robustly detected in the parapineal. Indeed, we rarely detected parapineal cells expressing GFP at 30hpf (n=3/19) and no expression was observed in the parapineal at 36hpf (n=0/10).

In parallel, we analyzed the expression of several known target genes of the Notch pathway, including *her2* (Quillien et al., 2011)*, her4* (Chapouton et al., 2011)*, her5 (*Ninkovic et al., 2005)*, her6* (Chapouton et al., 2011)*, her9* (Chapouton et al., 2011)*, her12* (Nikolaou et al., 2009)*, her15* (Webb et al., 2009). Among the *her* genes analyzed, we found that two of them, *her6* and *her9*, are robustly expressed in parapineal cells in a mosaic pattern (n=22 for *her6* and n=8 for *her9*), in addition to other brain regions (see Author response image 3). The expression of both *her6* and *her9* appear reduced in many brain regions of *mib-/-* mutants but, in the parapineal, we found that only the number of *her6* expressing cells (n=21) but not the number of *her9* expressing cells (n=10) was decreased (see Author response image 3). This result could be added in a revised version of our manuscript as a Figure supplement for Figure 1.

**Author response image 3. respfig3:** Notch target genes *her6* and *her9* are mosaïcally expressed in parapineal cells. (A-D) Confocal sections showing the expression of her6 (A, B) and her9 (C, D) (red) at 32 hpf in control embryos (A, C) and in mib-/- mutant embryos (B, D) merged with a cell nuclei staining (grey). (E-F) Dot plots showing the number of her6 (E) and her9 (F) in control (Blue marks) and mib-/- mutant embryos (Orange marks). Embryo view is dorsal, anterior is up; epiphysis (white circle), parapineal gland (yellow circle); scale bar=10 µm. Mean ± SEM are indicated as long and short bars. ** P-value<0.01; * P-value<0.05 in welsh t test (Wilcoxon test); data corresponds to one experiment.

3) While the study provides insights about when Notch activity is required, much less can be said about exactly where it is required. The authors can speculate in the discussion but must remain circumspect about this issue.

Our LY411575 treatment gives us a good idea of when Notch is required to control the laterality of parapineal migration (early time window 8-22 hpf) and the overall number of PP cells (22-32 hpf). Given that the LY411575 does not block migration in any of these time windows however, we agree that we cannot be fully sure about the timing of Notch requirement for migration *per se*; some possible reasons for this were already presented in the original discussion.

While our loss of function (LOF) experiments are not informative concerning migration, our gain of function (GOF) experiments strongly suggest that Notch acts at the time the PP initiates migration. As suggested by reviewer 1 in one of his/her specific comment, it is possible that NICD persists long after the heat shock and, in that case, Notch could act after 32 hpf. As the PP has usually migrated by 32 hpf, however, it is likely that the Notch pathway acts before this stage to explain the full PP migration defects observed in some *mib-/-* mutants. We nonetheless agree with Reviewer 1 that Notch could contribute to migration after 32hpf. To address this possibility, we treated embryos with LY411575 during a later (32-36 hpf) or extended time window (22-48 hpf). We did not observe any effect of these treatment on the migration as assessed by *gfi1a* expression at 48hpf (Author response image 4); likewise, we did not observe changes in the number of *gfi1a+* cells upon these late treatment.

**Author response image 4. respfig4:** Late LY41 treatment does not affect parapineal migration. (**A-B**) Upper panels show a schematic of the LY411575 treatment timeline, from 32 to 36 hfp (**A**) or 22 to 48 hpf (**B**) for dot plots. Dot plots showing, for each embryo, the mean position of *gfi1ab* expressing cells at 48 hpf in the parapineal of DMSO treated wild-type (*+/+,* light blue dots; A, n=12; B, n=15), DMSO treated *mib+/-* heterozygote (dark blue dots; A, n=15; B, n=17), LY411575 treated wild-type (light red dots; A, n=12; B, n=23) and LY411575 treated *mib+/-* embryos (dark red dots, A, n=13; B, n=17). No migration defect was observed in either context; data corresponds to one experiment.

We could modify the Discussion section of a revised manuscript to modulate our conclusion: whereas our original text read ‘The fact that ectopic Notch signaling at late stages (26-28 hpf) can efficiently block migration argues for a role of Notch at the time when the parapineal initiates its migration’, it could be replaced by ‘Given that LY411575 does not block migration in any of the tested time window, we cannot be absolutely sure about the timing of the Notch requirement for migration *per se*. However, our GOF experiments strongly suggest that Notch acts around the time that the PP initiates migration and not earlier in development.’

4) Although it would be interesting to know which Notch receptors and ligands are at play, this seems beyond the scope of what is necessary for this study.

We agree that it would be interesting to know whether component of the Notch pathways are indeed expressed in the PP during the appropriate time, and we had already begun to address this question over the course of our study. Among the genes encoding Notch ligands, we analyzed the expression of *deltaA* and *deltaB* in the epithalamus. We found that *deltaB* is expressed in one or two parapineal cells at both 28 hpf (n=6/12) and 32 hpf (n=17/23) (See Author response image 5); no expression could be detected in the PP of the remaining embryos (n=6/12 at 28 hpf or n=6/23 at 32 hpf), suggesting that *deltaB* expression is highly dynamic. We found that *deltaA* mRNA could also be detected at 32 hpf in 1-2 PP cell, although its expression is less robust than *deltaB* mRNA (n=3/10) (not shown).

**Author response image 5. respfig5:** Notch ligand *deltaB* is mosaically expressed in the parapineal. (**A-B’’**) Confocal sections showing the expression of *Tg(dusp6:d2EGFP)* transgene (green, **A, B**) or *deltaB* (red, **A’-B’**) and merge (**A’’-B’’**) in the epithalamia of two representative embryos at 32 hpf. *deltaB* mRNA is detected in only one or two parapineal cells in most embryos (n=17/23); staining is either co-expressed with *Tg(dusp6:d2EGFP)* (n=11/23) or detected in *Tg(dusp6:d2EGFP)* negative cells (n=6/23); in n=6/23, *deltaB* was not detected in the parapineal. Embryo view is dorsal, anterior is up; epiphysis (white circle), parapineal gland (yellow circle); scale bar=10 µm. Data representative of 2 experiments.

When detected in the PP, *deltaB* expression was found to overlap with *Tg(dusp6:d2EGFP)* expression in between half (n=3/6 at 28hpf) to two thirds of the embryos (n=11/17 at 32 hpf). If a lateral inhibition model is at play downstream of Notch signaling to restrict FGF pathway, we expected to find perfect co-expression of at least one Delta with the *Tg(dusp6:d2EGFP)* transgene, and non overlapping expression of Notch targets, such as the *her* genes. Although *deltaB* expression is most often found co-expressed with *Tg(dusp6:d2EGFP)* (in about 2/3 of the embryos at 32hpf), this is not always the case. Similarly, the expression of *her6* and *her9* can sometime overlap with *Tg(dusp6:d2EGFP)* expressing cells. These results might be explained by the differential dynamic that exist in the detection of a GFP transgene versus mRNA. Alternatively, the classical lateral inhibition model might be too simplistic to recapitulate the complexity of interactions between the Notch and FGF pathways.

Altogether, our results show that both components and read-outs of the Notch pathway can be detected mosaically in the PP, supporting a role of Notch signaling within the PP in restricting FGF activation to a few cells. We agree with reviewer 2 and the Editor that addressing further the role of Notch components and target genes is beyond the scope of our study and this is the reason why these data were not included in our original manuscript. However, as suggested for *her6* and *her9* Figure, we could easily add our data on *deltaB* expression in a future supplement Figure if the reviewers and the editor think it would be pertinent.

Below are the full reviews from the three reviewers which we hope you find helpful.Reviewer #1:The manuscript describes how Notch signaling affects parapineal cell migration and cell type specification in the zebrafish brain. The manuscript is clearly written and illustrated, and the Discussion section is informative and thoughtful. The authors demonstrate that loss of Notch signaling leads to an expansion of the FGF pathway and defects in parapineal migration and laterality, and an increase in the number of gfi1ab and sox1a expressing parapineal cells. They further demonstrate that the roles of Notch signaling in cell type specification, cell migration and laterality of cell migration are uncoupled. Only the migration and laterality defects are FGF dependent, whereas the effect of Notch signaling on cell type specification is not. The authors also show that these different functions of Notch signaling occur at different time points of development. Cell migration to the left side (laterality) depends on Notch signaling before migration starts (8-22 hpf) by restricting Nodal signaling to the left side of the epithalamus. Whereas cell type specification occurs later. The main conclusion of the manuscript is that proper parapineal cell migration depends on Notch-dependent restriction of FGF signaling to just a few tip cells. The mechanisms by which Notch restrict FGF signaling is unknown.The results are clearly illustrated and the majority of the data interpretation are sound.

We thank reviewer 1 for his/her positive comments.

However, the manuscript would profit from a more detailed description of the different cell types.

As it is a small structure, about 15 cells, cell diversity in the parapineal has probably been underestimated until now; in previous papers, it has always been assumed that all described markers label the same cells in the PP. Our results showing a difference in the behavior of *tbx2b+* PP cells versus *sox1a* and *gfi1a+* cells suggest that the pool of *tbx2b+* parapineal cells differs from a *sox1a/gfi1a+* pool. We have preliminary data, by double fluorescent in situ, confirming that *tbx2b* and *gfi1a* indeed label two distinct pools of PP cells. However, we are reticent to include these data in our manuscript as our study mainly focuses on the role of Notch signaling in controlling the collective migration of parapineal cells through its capacity to restrict FGF pathway activation to a few leading cells and not the potential changes to different PP cell types. In that context, we feel that adding data on the characterization of PP cells subtypes might dilute our main message.

Also, the fact that suboptimal doses of SU5402 rescue the mib migration phenotype should be discussed/analyzed in more detail, as the results might suggest that it is not important that only the tip cells express FGF signaling but rather that these tip cells express the correct level of FGF signaling.

See our reply concerning the major issues raised.

Also, there is no description of where Notch signaling is active. As Notch reporter lines exist, the authors should add this information and discuss it.

See our reply concerning the major issues raised.

Specific comments:1) Subsection “The parapineal of mindbomb mutant embryos display expanded FGF pathway activation”: The number of parapineal cells is slightly increased but the number of sox1a positive cells is not increased (which is a marker for parapineal cells). Are there sox1a-negative parapineal cells or is the difference between Figure 1F and 1G due to biological noise?

Yes, some cells are negative for *sox1a* but seems to be part of the rosette highlighted by staining nuclei.

2) Figure 1A: please indicate in the figure legend or figure what structures are outlined by yellow and white dotted lines.

We apologize for this oversight and will add this information in the Figure.1 legend.

3) Subsection “Loss of Notch signaling results in an increase in the number of certain parapineal cell subtypes”: A more detailed characterization of tbx2b, sox1a and gfi1a expressing cells is necessary. Are sox1a and gfi1ab expressed in the same cells?

See our reply above.

4) Subsection “Loss of Notch signaling results in an increase in the number of certain parapineal cell subtypes” (and subsection “Notch effects on parapineal cell specification can be uncoupled from migration”) the authors conclude that Notch probably controls the transition from tbx2b-expressing putative progenitors to sox1a/gfi1ab-expressing differentiated cells. After loss of Notch signaling in mib mutants, the number of sox1a/gfi1ab positive cells increases, whereas the tbx2b positive cell number remains the same. How do the authors envision that the progenitor pool stays the same? Normally, defects in Notch dependent lateral inhibition should lead to the increase of one cell population at the expense of another. Do the progenitor cells divide asymmetrically and only the future sox1a/gfi1ab cells proliferate more? In this scenario tbx2b-expressing progenitor cells are independent of Notch signaling. In many other systems progenitor maintenance crucially depends on Notch signaling. Either the parapineal is different or the tbx2b positive cells are not self-renewing progenitors? Please discuss in more detail what is known about tbx2b, sox1a and gfi1ab positive cells.

Nothing is known about the lineage of PP cells. This question is indeed interesting, but we have not addressed it further as it is not the main message of our paper.

5) Regarding the hs:NICD experiments: Are control embryos also heat shocked? Please label figure accordingly.

Yes, control embryos are non-transgenic or single *Tg(UAS:NICD)* or *Tg(hsp70:Gal4)* transgenic embryos that were heat-shocked under the same conditions as the double transgenic embryos. We can mention this in the Materials and methods section or in the Figure Legend of a revised version of our manuscript.

6) Figure 2I: The time-line of treatment shown above the graph is very useful. It would facilitate the interpretation even further if it was indicated at what time point (32 hpf) migration begins.

The initiation of parapineal migration is variable but usually starts between 28hpf and 30 hpf; at 32 hpf parapineal position is obviously displaced from the midline. We can add a migration box (28-30hpf) on the time-line.

7) The fact that ectopic Notch signaling at late stages (26-28 hpf) can efficiently block migration argues for a role of Notch at the time when the parapineal initiates its migration. If Notch signaling is indeed required just before migration, it is unclear why LY411575 treatment (22 to 32 hpf) does not affect migration. Maybe activation of Notch by inducing NICD until 28hpf persists much longer and NICD is still active after 28hpf. Therefore, embryos should be treated with LY411575 beyond 32hpf, as Notch might be required for migration during these later stages.

See our reply concerning the major issues raised and Author response image 4.

In our original discussion, we suggest that LY411575 might not completely block Notch signaling and that the differential effect of LY411575 on specification and migration of parapineal cells could reflect a different requirement in Notch signaling threshold. To keep our original manuscript short, we did not include other speculative hypotheses.

8) Subsection “Notch activity is required early for unilateral activation of the Nodal pathway in the epithalamus”: 'To determine whether the parapineal migration defects we observed (add: in mib mutants) might be a direct…. In sentence in the paragraph above the authors conclude that there was no migration defect in LY411575 treated embryos, which is confusing.

Indeed, although LY411575 treatment from 22 to 32hpf promotes an increase in the number of *gfi1ab* positive parapineal cells, it does not affect parapineal migration. As suggested by the reviewer, we can add ‘in *mib-/-* mutants and *rbpj* morphant embryos’ in the previous sentence to remind the reader that PP migration defects were observed in these two contexts only.

9) Subsection “Decreasing FGF signaling rescues the parapineal migration defects in loss of Notch context while increasing FGF signaling aggravates it”. The authors show that suboptimal doses of the FGF inhibitor SU5402 rescue to some degree the migration defect in mib mutants, which are characterized by FGF signaling activation in all migrating cells. This experiment actually argues that it is not important that the tip cells possess more FGF signaling then the follower cells but rather that FGF signaling needs to be expressed at the correct level. The authors should discuss the results of this experiment in more detail.

We have replied to this point above. We can add a sentence in the discussion to mention this alternative hypothesis.

10) Subsection “Decreasing FGF signaling rescues the parapineal migration defects in loss of Notch context while increasing FGF signaling aggravates it”: 'Taken together, our.…experiments argue that the Notch pathway is required for parapineal migration, and that it acts by restricting FGF activation in parapineal cells.' However, in subsection “Notch activity is required early for unilateral activation of the Nodal pathway in the epithalamus” the authors argue that Notch and FGF act synergistically during migration, as rbpj morphant:cafgfr1a embryos display a more severe migration phenotype then CAfgfr1 embryos alone. Therefore, one has to conclude that Notch and FGF partially act in parallel on parapineal migration, correct?

By using ‘synergistic’, our goal was not to suggest that FGF and Notch act in parallel but rather that the two pathways appear to interact. Indeed, our data suggest that Notch acts upstream of the FGF pathway as LOF and GOF for Notch trigger an increase or decrease in FGF signaling in the PP. However, we understand from reviewer 1's comment that the wording chosen might be confusing. We can change the sentence ‘The interaction detected… ‘to ‘Notch and FGF pathways appears to interact in this context as the increase in parapineal migration defects observed in *rbpja/b* morphant embryos expressing the activated FGF receptor is significantly higher than expected from simply adding the effects of activated receptor transgene and *rbpja/b* MO injections alone.’

11) Epistasis: Notch and FGF could also act in a feedback loop. Is Notch signaling affected after manipulation of FGF signaling?

It is an interesting possibility. We could check *her9* and *her6* expression in *fgf8* mutants or in SU5402 treated embryos. This said however, we feel it might dilute our message as this question is beyond the main focus of our study.

Reviewer #2:The paper by Lu et al., is an excellent follow up on previous work from Roussigne which demonstrated a role for restricted FGF signaling in collective migration of the parapineal cells. In this study the authors make good use of a combination of transgenic tools, mutants and chemical inhibitors to demonstrate the role of Notch signaling in a particular time window in restricting FGF signaling to a limited number of parapineal cells to determine asymmetric collective migration. They show that it is required after a relatively early stage when Notch is required to determine asymmetric Nodal signaling and after a stage is required to restrict the number of cells with gf1ab and sox1a positive cells. The experiments are presented in a logical order, the data is presented well, and its analysis supports the conclusions of the authors.I have no major problems with the paper in its current form. My issues were only tangentially related to the study.

We thank reviewer 2 for his enthusiasm.

At various points in the Introduction and Discussion section the authors make comparisons with the role of FGF and Notch signaling in the lateral line system. I found those references confusing and potentially misleading. I would have liked to hear more about the role of Notch signaling in restricting migratory potential in tracheal cells and endothelial cells in vascular development. While the tracheal and vascular systems are referenced, the comparison could be elaborated on.

The interaction between Notch and FGF/RTK pathways in the tracheal and vascular systems is not well understood but we could discuss it more if required.

In the Introduction the authors suggest that the Dalle Nogare paper shows a role for FGF signaling in maintaining cluster cohesion. It is not clear what the authors have in mind here because the primary role of FGF shown in that paper is to suggest a role for FGFs released by leading cells in providing a guidance cue for collective migration of trailing cells. Their description of FGFs role in cluster cohesion is not essential for any point they wish to establish in this study and may only serve to establish a misleading impression about conclusions in the Dalle Nogare paper in the minds of readers for those not familiar with that paper.

By writing ‘a role of FGF in cluster cohesion’, we were referring to a function of FGF leading to trailing signaling in preventing splitting of the primordium. Therefore, we fully agree with reviewer 2 on the interpretation of the data from Dalle Nogare et al. We understand that referring to ‘cluster cohesion’ to summarize these data is misleading and we can re-word this sentence.

Similarly, the authors contrast the role of Notch, downstream of FGF, in determining apical constriction in forming neuromasts in the migrating primordium, with the role of Notch in restricting the number of cells with FGF activity in the context of parapineal migration. The authors are not wrong here but they may give readers that not familiar with the Lateral Line system a simplistic idea about the role of Notch signaling in the primordium, where, as in the parapineal, it has multiple sequential roles. For example, first in restricting sensory hair cell progenitor fate to a central cell, and subsequently in determining Notch activation in neighboring cells to consolidate morphogenesis of epithelial rosettes. It is true that activation of Notch promotes apical constriction in the absence of FGF signaling and therefore in this context, downstream of FGF signaling and different in its epistatic relationship from the role of Notch in the parapineal. However, in the context of lateral inhibition and in restricting the number of cells with a particular fate/activity there may be important similarities in the role of Notch in the lateral line primordium and in parapineal cells. Without such a clarification, I felt the contrasting of the role of Notch in the parapineal and primordium could be potentially misleading.

To keep the manuscript‘s length within reason, we indeed decided to focus our discussion on the differences that exist in the interaction between Notch and FGF signaling in the PP (Notch upstream FGF) versus the LLP (FGF upstream Notch) rather than also including text on the role of Notch alone. We agree that, in both models (PP and LLP), the Notch pathway plays a similar role in restricting the number of differentiated cells. Nonetheless, we don’t feel we have the space to describe in detail the LLP system. We can add a sentence to mention the existence of similarities in the role of Notch between these two models, as suggested by reviewer 2 if requested.

A question that remained unanswered, which I was curious about and is not necessary in my mind for acceptance of the paper- which cells does Notch activity required in for its role in restricting migratory potential, which Notch ligands and receptors are required for this process and does their spatial distribution or that of downstream target genes provide insight about where Notch activity if required for restriction of FGF signaling in the parapineal.

See our reply above and Author response image 3 and Author response image 5.

Reviewer #3:In this study by Wei et al., the authors investigate collective cell migration of parapineal cells during development using zebrafish as a model system. The authors found that global inhibition of Notch signaling, either by a chemical inhibitor or a MO-mediated knockdown, increased the number of cells that activate FGF signaling. As FGF is required for the parapineal migration and it is typically restricted to a few leading cells, this blocked the parapineal migration. Conversely, upregulating Notch signaling led to the loss of FGF signaling and also defects in its migration. These manipulations also affected a number of parapineal cells. Finally, the authors modulated FGF signaling in both Notch loss- and gain-of-function conditions. Decreasing FGF under this condition resulted in a partial rescue of the migration, whereas gain of FGF signaling worsened the migration phenotype. Interestingly, these manipulations did not affect the number of parapineal cells, indicating that these two roles of Notch signaling are distinct. Based on these data, the authors concluded that Notch restricts FGF signaling to a few leading parapineal cells, a process required for the proper migration. In addition, during later stages, Notch signaling controls the number of parapineal cells and this particular role of Notch can be uncoupled from its earlier role in restricting FGF signaling.Overall, the conclusions are based on carefully done and controlled experiments (with a few exceptions). And while they provide some interesting insights into mechanisms of parapineal migration, the study is quite descriptive and does not go beyond the superficial analysis of cellular phenotypes. It is also not clear which cells express Notch ligands and receptors and whether effects on FGF signaling are direct. In summary, the overall findings are quite specialized and somewhat incremental and, thus, more suited for a specialized audience.

We thank reviewer 3 for acknowledging the quality of our work but we disagree with him/her on the fact that our findings will be more suited for specialized journal.

As mentioned in our cover letter to the editor, our work will be of interest to the community working on left-right brain asymmetry as it sets a framework to address how Nodal signaling provides a leftward bias to the Notch dependent restriction of FGF signaling that is required for parapineal laterality and subsequent habenular asymmetries. However, we believe that our study will also be of interest to groups interested in collective cell migration in general. Indeed, our findings potentially apply to other models of FGF dependent cell migration and, in the future, they would set the stage for studies investigating how Notch can restrict the activation of FGF pathway and how other factors or signaling pathways modulate this Notch dependent restriction in normal or pathological contexts of collective cell migration, including metastasis. For these reasons, we believe that our study will be of considerable interest to the broad audience readership of *eLife*.

Major comments:Subsection “Loss of Notch signaling results in defects in parapineal migration”: Indicates that migration of the parapineal failed in 1/3 of mib-/-. I don't believe the authors discussed why this is the case/why the phenotype is partially penetrant in mib mutants? As well as why this phenotype was partially penetrant in the rbpj MO-injected embryos (with only 13% failing to migrate).

Why a single gene mutation results in a partially penetrant phenotype (expressivity) is a common question in genetics, and probably depend on various parameters such as individual variation in the genetic background, a redundancy of the mechanisms at play or intrinsic properties of the system that make it more or less robust.

In the case of parapineal migration, the number of *Tg(dusp6:d2EGFP)+* cells is quite variable suggesting that parapineal migration tolerates variation in the degree of mosaicism of FGF pathway activation and/or in the level of FGF signaling. The general robustness of parapineal migration might also reflect that migration is very short in terms of distance travelled (measured in tens of µm) compared to, for instance, the migration of the lateral line primordium (LLP; measured in many hundreds of µm). As a consequence of the scale, intermediate migration phenotypes that can be observed for LLP migration might not be easily visible for PP migration. Indeed, in our study, we chose to define a parapineal as non-migrating when its mean position was found between -15 μm and +15 μm of the midline (the standard width of the epiphysis). One can notice, however, that in the *mib-/-* mutant embryos where the parapineal migrates further than -15µm, the parapineal often migrates less far from the midline than in wildtype (Figure 1N); we thus do observe a range of phenotypes that goes from no migration to delayed migration rather than a black or white phenotype.

rbpj a/b MO: given specific guidelines for MOs (Stainier et al., 2017), these experiments do not seem to be appropriately controlled.

These MO have been previously validated as their injection phenocopy the *mib-/-* mutant phenotype in neurogenesis and somites boundary (Echeverri and Oates, 2007). Furthermore, in our study it is only used as a second approach to confirm a role of the Notch pathway in parallel to our use of *midbomb* mutants. As the phenotypes are similar, we do not see a need to control as strictly as if we were only using the morpholinos.

The results related to the inhibitor treatment seem odd. If the γ-secretase inhibitor, LY411575, is specific and Notch is playing a role in migration and specification of this system, I would expect to see a result with treatment from the 22-32 hpf time period. This drug and it has only been published in two zebrafish papers. Why not use the γ-secretase inhibitor DAPT that has been utilized in many other studies. Related to this, I am confused that 8-22 hour inhibitor treatment did not interfere with the Tg(dusp6:d2EGFP) transgene expression, neither did the 22-32 hour treatment. What is then the time window for Notch signaling that controls Tg(dusp6:d2EGFP) expression?

We decided to use LY411575 as it is understood to be more powerful compared to the previously described γ-secretase inhibitor DAPT. For instance, in Rothenaigner et al., 2011, the authors compared the γ-secretase inhibitor DAPT with a dose range of its second-generation derivative, LY411575 and noted that ‘LY411575 (10 µM) proved as efficient as 100 µM DAPT in interfering with Notch signaling. As such, since its first description in 2007 (Fauq et al., 2007) and its first use in zebrafish (Rothenaigner et al., 2011), LY411575 has become the gold standard in the literature at the expense of DAPT; although numbers are difficult to assess, there are definitively more studies describing the use of LY411575 in zebrafish than just the two papers one can find referenced on Pubmed when using ‘zebrafish’ and ‘LY411575’ as searching keywords. For instance, recently, Kozlovskaja-Gumbrienė et al., (*eLife* 2017) or Thomas et al., (*eLife* 2019) used LY411575 to block Notch pathway in the migrating lateral line primordium or in neuromasts and these studies are not among the publication easily found in Pubmed with searching keywords.

Concerning the second point, treating with LY411575 between 8-22 hpf or 22-32 hpf does not block parapineal migration and we speculated in the discussion about potential reasons. However, the fact that normal migration correlates with no significant changes in the pattern of the *Tg(dusp6:d2EGFP)* FGF reporter transgene for both windows treatment support the conclusion of our study.

Throughout the manuscript, it is not clear where Notch signaling is active/ where Notch signaling components are expressed. It would be helpful to see the expression of where particular Notch signaling components are expressed during the time points of interest to further validate the reason behind studying Notch signaling. In other words, is Notch acting directly on parapineal cells?

See our reply concerning the major issues raised and Figure 3 and Figure 5 for reviewers.

[Editors’ notes: the authors’ response after being formally invited to submit a revised submission follows.]

Further to my previous email, the editors and reviewers have considered your plan and invite you to proceed with your revisions as proposed. The asked us to pass on the following comments:The authors should expand their discussion, add the discussion points raised by the reviewers and add relevant figure panels.

As mentioned above, we have added 3 Figure supplements and discussed them to address the concerns raised by the reviewers. We have also made the changes to the Introduction, Results section, Discussion section, Materials and Methods section and Figure legends requested by the reviewers (See at the end below).

Specific comments:In Figure 3 for reviewers they should add panels that only show the red signal.

As our Author response image 3 was based on a single experiment, we performed 2 more experiments to confirm *her6* and *her9* expression in the parapineal and the decrease of *her6* in the parapineal of *mib^-/-^*mutants. The expression of *her6* and *her9* appear to be globally decreased in *mib^-/-^* mutants, as we observed previously, but also in LY411575 treated embryos. While we have managed to confirm the expression of both of these genes in the parapineal, we could not confirm a significant decrease in the number of *her6+* cells in the parapineal of *mib^-/-^* mutants, despite the high number of embryos analyzed (n=43 controls and n=41 *mib^-/-^* embryos). One explanation is probably that fluorescent *in situ* hybridization are not quantitative enough to allow us to detect modest decreases in the level of expression or in the number of *her* expressing parapineal cells. In addition, *her6* and *her9* expression as detected by fluorescent *in situ* hybridizations is often weak and/or punctate, which makes the quantification difficult. While it remains possible that the expression of both *her6* and/or *her9* are targets of Notch pathway in the parapineal, as they appear to be elsewhere, we haven't managed to reproduce our initial results and have decided not to include the expression of *her6* and *her9* in our revised manuscript.

We have, on the other hand, included a Figure1—figure supplement 3 showing the expression of *deltaB* that supports the existence of active Notch signaling in the parapineal. We have also included 2 examples of embryos with *Tg(Tp1:GFP)* Notch reporter expression detected in the parapineal (n=3/19). The reason for the lack of robustness of expression of the Notch reporter line is unclear. Also, as *Tg(Tp1:GFP)* expression was too rarely detected in the parapineal, we have not analyzed it in *mib^-/-^* mutants or LY411575 treated embryos.

As LY411575 does not inhibit migration at any time point, the authors should test if this drug is able to completely reduce Notch signaling. Is the signal in the TP Notch reporter line gone after treatment? Possibly, a small amount of Notch signaling is sufficient to allow for migration.

As mentioned above and in the revised version of manuscript presented here, we did not detect robust expression of the *Tg(Tp1:EGFP)* reporter line in the parapineal. As such, we have not analyzed *Tg(Tp1:GFP)* expression in LY411575 treated embryos. To test our LY411575 treatment regime, we chose to analyze the expression of *her6* and *her9* (expressed in the parapineal of most embryos) and *her4*, a well-described consensus Notch target gene. We found that *her4* expression is completely lost in both *mib^-/-^* mutants and upon LY411575 treatment (100µM from 22 to 32 hpf); see below our Figure for Reviewers. We also observed a global decrease in the expression of *her6* and a slight reduction of *her9* expression in LY411575 treated embryos as seen in *mib^-/-^* mutant embryos. Our data confirm previous work showing that the abundance of *her6* transcripts is significantly reduced, but not lost, in the dorsal diencephalon and hindbrain of *mib^-/-^*mutants (Cunliffe, 2004). Previous studies have also already suggested that late *her9* expression would be partly dependent on Notch signaling (while its early expression is not) (Bae, 2005). Thus, while our LY411575 treatment does not affect parapineal migration, it appears to block or reduce the expression of the 3 tested *her* genes as previously described in *mib^-/-^* mutants.

The expression data of Notch pathway members should be included and discussed in the manuscript.

As mentioned above, we have not included the expression of *her6* and *her9* in parapineal cells as we haven't managed to detect a robust decrease of their expression in *mib^-/-^* mutants and, thus, cannot confirm that they are Notch pathway components in the context of the parapineal. In the new version of our manuscript, however, we have included the expression data for *deltaB*, which can be found in one or two parapineal cells in about half of the embryos analyzed (Figure 1—figure supplement 3).

The text describing the different cell types should be clarified, as it is currently unclear which genes mark possibly the same cells or which genes label all cells.

As mentioned in our previous reply, parapineal cells diversity has yet to be thoroughly characterized in the literature, with all parapineal markers assumed to label all parapineal cells. Our data suggest that the pool of *tbx2b+* parapineal cells differs from a *sox1a/gfi1a+* pool. Our preliminary data from double fluorescent in situs shows that *tbx2b* and *gfi1a* indeed label two distinct pools of cells. However, while these findings are interesting, we fear that adding data on the characterization of parapineal cell subtypes might dilute the main message of our study that focuses on the role of Notch signaling in controlling the collective migration of parapineal cells through a restriction of FGF pathway activation. In our revised manuscript, we have nonetheless added/modified the text to clarify what is described in the literature and what is suggested by our study. This includes:

Subsection “Loss of Notch signaling results in defects in parapineal migration”: We added the sentence “We confirmed this result by using another parapineal specific marker, *gfi1ab* (Dufourcq et al., 2004), whose expression is detected at a later stage (from 36-40 hpf) than *sox1a* (from 28 hpf) in parapineal cells.”

Subsection “Loss of Notch signaling results in an increase in the number of certain parapineal cell subtypes”: We modified the sentence to read “In contrast, the number of parapineal cells expressing *tbx2b*, a parapineal marker previously suggested to be required for the specification of parapineal cells (Snelson et al., 2008), was not increased in *mib*^-/-^ mutants (Figure 1J-1K, 1Q)."

Subsection “Notch effects on parapineal cell specification can be uncoupled from migration”: We added the sentence “Therefore, while in previous studies it was assumed that all described parapineal markers (*tbx2b, sox1a, gfi1a*b) label the same cells, our data indicate that the pool of *tbx2b+* parapineal cells probably differs from a *sox1a/gfi1a+* pool.”

[Editors' note: further revisions were requested prior to acceptance, as described below.]

Thank you for submitting your article "Notch signaling restricts FGF pathway activation in parapineal cells to promote their collective migration" for consideration by eLife. Your article has been reviewed by Marianne Bronner as the Senior Editor, a Reviewing Editor, and two reviewers. The reviewers have opted to remain anonymous.The reviewers have discussed the reviews with one another and the Reviewing Editor has drafted this decision to help you prepare a revised submission. Please aim to submit the revised version within two months.The reviewers felt that manuscript is much improved and addressed the main concerns to: (1) provide evidence that some Notch pathway signaling members are expressed in the parapineal during migration; (2) further investigate effects of the γ-secretase inhibitor treatment on the Notch pathway activation, and (3) clarify the dynamics of parapineal marker gene expression during PP development. However, the evidence that Notch pathway acts within parapineal cells remains weak: deltaB and the reporter expression are not robust enough to make that conclusion, and functional experiments testing this hypothesis are absent. Therefore, please tone down that conclusions and present alternative possibilities.We will look forward to hearing from you with a revised article with tracked changes, and a response letter (uploaded as an editable file) describing the changes made in response to the decision and review comments.

We are grateful that the reviewers found our revised manuscript is an improvement and we would like to thank them again for their constructive comments.

Concerning the final request of the reviewers, as we detect mosaic expression of *deltaB* Notch ligand in the parapineal of a majority of embryos (1 or 2 cells in more than 2/3 of embryos at 32 hpf) and the expression of some Notch target genes (*her6* and *her9*), our preferred hypothesis is that Notch signaling acts in the parapineal to restrict FGF pathway. However, we fully agree that we have not provided concrete proof of this, and have moderated our conclusion accordingly. For this, we have added sentences to our discussion mentioning that Notch signaling could be required in the tissue surrounding the parapineal rather than in the parapineal itself or could possibly act simultaneously within and outside parapineal cells. We believe that our modifications correspond to your expectations but would be happy to consider any specific suggestions to improve further this point.